# CAPE: Encoding Relative Positions with Continuous Augmented Positional Embeddings

**Tatiana Likhomanenko**[*]
Facebook AI Research
tata.antares@gmail.com

**Qiantong Xu**
Facebook AI Research
qiantong@fb.com

**Gabriel Synnaeve**
Facebook AI Research
gab@fb.com

**Ronan Collobert**
Facebook AI Research
locronan@fb.com

**Alex Rogozhnikov**
Herophilus, Inc.
alex.rogozhnikov@yandex.ru

## Abstract

Without positional information, attention-based Transformer neural networks are permutation-invariant. Absolute or relative positional embeddings are the most popular ways to feed Transformer models with positional information. Absolute positional embeddings are simple to implement, but suffer from generalization issues when evaluating on sequences longer than seen at training time. Relative positions are more robust to input length change, but are more complex to implement and yield inferior model throughput due to extra computational and memory costs. In this paper, we propose an augmentation-based approach (CAPE) for absolute positional embeddings, which keeps the advantages of both absolute (simplicity and speed) and relative positional embeddings (better generalization). In addition, our empirical evaluation on state-of-the-art models in machine translation, image and speech recognition demonstrates that CAPE leads to better generalization performance as well as increased stability with respect to training hyper-parameters.

## 1 Introduction

Transformers have been shown to be highly effective on problems involving sequential modeling, such as machine translation (MT) [48] and natural language processing (NLP) [14, 41, 6]. Following its success on these tasks, the Transformer architecture raised immediate interest in other domains: automatic speech recognition (ASR) [15, 23], music generation [26], object detection [7], and finally image recognition [16, 47] and video understanding [5].

Two major components of the Transformer are the attention mechanism [2, 48] and the positional encoding [48, 44, 26, 11]. Without the latter, vanilla attention Transformers are invariant with respect to input tokens permutations (making "cat eats fish" and "fish eats cat" identical to the model). In the original Transformer publication, sinusoidal positional encoding was introduced [48]. Token positions were encoded in an absolute manner, which was sufficient to achieve state-of-the-art performance in numerous tasks. However, performance issues were later observed when dealing with sequences of length not seen at training time [26, 11, 59, 33, 42, 18]. For most applications relative positions between tokens are more relevant than absolute ones. A number of approaches were thus proposed to encode relative positions in an explicit form [44, 11, 26, 42, 27], leading to improvements in modeling long sequences. However, all these approaches focus on modifying the attention mechanism and suffer from additional computational and memory costs [57]. Relative positional encoding is also

---

[*]Currently at Apple.

notably hard to implement efficiently for multidimensional case, and recent advances in Transformer models for computer vision [7, 16, 47] still rely on learnable absolute positional encoding.

Instead of changing the Transformer attention mechanism, we propose to improve absolute sinusoidal positional encodings in two ways: a) instead of discrete positions, rely on continuous ones, which better match the continuous nature of image, sound or video data; b) preserve some information about relative token positions, via a specific augmentation approach for positional embeddings during training. We empirically evaluate our approach, dubbed continuous augmented positional embedding (CAPE), with recent state-of-the-art models in several application domains. We study generalization properties and introduce unique features unlocked by CAPE. The main contributions of this work are:

- new augmented continuous positional embedding (CAPE), which encodes some relative position information in a computationally efficient way, and improves generalization performance compared to other positional embeddings across a variety of domains: machine translation, image and speech recognition;

- a single vision Transformer (UniViT) trained with CAPE on the mix of different resolutions: it outperforms each single-resolution baseline, generalizes better to unseen resolutions and can naturally process images of any size;

- new CAPE-based adaptive training scheme for ASR that eliminates need for padding.

## 2   Related Works

Since the appearance of Transformers, many works have investigated ways to encode positional information. A detailed analysis of various positional embeddings is available for BERT architecture [49], where authors empirically relate properties of positional embeddings to performance on downstream NLP tasks. A recent study [28] (also focuses on BERT) highlights the negative impact of spurious correlations between word and positional embeddings, and proposes to explicitly disentangle the contribution of positional and content embeddings in the attention mechanism. In contrast, our approach implicitly enforces this disentanglement by leveraging augmentation.

Systematic studies of positional embeddings in audio domain are scarce. Several ways to encode relative positions for Transformer-based speech recognition are compared in [51]. Experiments show that absolute sinusoidal positional embeddings work no better than stacking consecutive frames at each time position (a particular form of convolution). We provide a more thorough evaluation of positional embeddings in ASR, over multiple datasets. We also show that embeddings obtained from a one-layer convolutional frontend benefits from adding positional information.

Transformers for computer vision applications are still in their early days, and most works rely on learnable absolute positional embeddings only [16, 47, 5, 1]. Several recent works complement the Transformer architecture with convolutional layers to induce a spacial relationship between tokens [21, 9, 53]. As we discuss later, this restricts flexibility compared to pure attention-based models. The work [21] suggests injecting learnable attention biases as an alternative mechanism to positional encoding. Evaluation of several positional encodings and their corresponding generalization has been done in a study [9] which is in line with our work. Convolutional elements were introduced in the Transformer architecture, leading to better generalization properties. In contrast, our experiments demonstrate that generalization can be achieved without architecture modification or convolutional inductive bias. Concerning video understanding, an evaluation of the impact of positional encoding was done in [34]: according to the results, positional encoding-free architectures perform best. Other work [5] reports that adding absolute positional embeddings improves models performance, but contribution of encoding space and time vary between datasets.

As a summary, many positional embeddings variants were previously introduced, often modality-specific. In our cross-modal study we focus on generalization properties of popular and widely used embeddings, and improve on absolute sinusoidal positional embedding, leading to a flexible Transformer architecture, with great generalization properties across a number of different tasks.

The closest idea to our work is augmentation of positions in Universal Transformer [12] where *discrete* global shifts are applied to synthetic tasks for encoder-decoder Transformer models. In our work, we introduce *continuous* augmentations, not *discrete*, which are more natural for continuous modalities like images and speech where CAPE benefits most and which are not discussed in [12].

In addition to global shifts our augmentations include global scaling and local shifts, and we also synchronize augmentations between encoder and decoder (Section 5.3).

## 3  Theoretical Analysis of Absolute Sinusoidal Positional Embeddings

Originally absolute positional $K$-dimensional embeddings were introduced in [48] as

$$E_{2k}(n) = \cos \omega_k n \qquad E_{2k+1}(n) = \sin \omega_k n \qquad \omega_k = 10000^{-2k/K} \qquad n \in \mathbb{Z}^+ \qquad (1)$$

with $k = 1, 2, \ldots, K/2$ enumerating components for a token at position $n$. For simplicity of analysis, we rewrite Eq. (1) as a complex-valued embeddings[2] with half the number of components:

$$\{\mathbf{E}(n)\}_k = E_k(n) = e^{i\omega_k n}$$

This definition can be rewritten in a recursive manner, by introducing a unitary operator $S$:

$$\mathbf{E}(n+1) = S\,\mathbf{E}(n), \qquad \text{with} \qquad \{S\,\mathbf{X}\}_k = X_k e^{i\omega_k} \qquad (2)$$

Therefore, the embedding at position $n$ contains sufficient information to compute the embedding of the next or previous positions, as applying $S^m$ ($m \in \mathbb{Z}$) performs a relative shift: $\mathbf{E}(n+m) = S^m\,\mathbf{E}(n)$. Variation in $\omega_i$ ensures that no positions $< 10^4$ are assigned similar embeddings. Before introducing augmentations of positional embeddings, we revisit positions parametrizations for different modalities.

**Positional encoding for text**  For natural language processing it is common to split text into words, letters, syllables and other sub-units. Original absolute sinusoidal positional embeddings enumerate these sub-units by their ordinal number $n$, a common choice that we follow.

**Positional encoding for images**  In a framework where patch and image sizes may vary, we find that enumerating patches is not appropriate, as positional embeddings may greatly differ for different scales of an image, leading to generalization issues. In that perspective, we consider scaled coordinates $x$ and $y$ that span interval $[-1, +1]$[3]. While previous works [16, 47] relied on learnable absolute positional embedding, we introduce the following $K$-dimensional absolute sinusoidal positional embedding defined for each position $(x, y) \in \mathbb{R}^2$:

$$E_{2k}(x, y) = \cos \pi (w_{k,x} x + w_{k,y} y) \qquad E_{2k+1}(x, y) = \sin \pi (w_{k,x} x + w_{k,y} y) \qquad (3)$$

$$w_{k,x} = 10^{2k/K} \cos k \qquad w_{k,y} = 10^{2k/K} \sin k \qquad (4)$$

Following Eq. (2), this corresponds to introducing two commuting unitary operators $S_x$ and $S_y$, for each unit shift in either direction on the plane. The choice of $w_{k,x}$ and $w_{k,y}$ is kept simple and deterministic, while giving no preference to any direction on the plane and providing angle of "hatching" (all components have different angle and angles uniformly cover possible directions), see Figure 9. Furthermore, embeddings defined by Eq. (3,4) allow attending to a specific small region around a point on the image without emphasizing points with only same $x$ or only same $y$ while we expect this artifact to contribute in case of concatenation of 1D embeddings for each axis [9, 16, 3, 38].

**Positional encoding for sound**  We propose to tie positional embeddings to timestamps in seconds. The embedding for a frame centered at $t$ seconds is given by:

$$E_{2k}(t) = \cos \omega_k t \qquad E_{2k+1}(t) = \sin \omega_k t \qquad \omega_k = 30 \times 10000^{-2k/K}$$

The choice of $\omega_k$ corresponds to the scaled version of Eq. (1) and ensures that queries with 30ms specificity are possible even with minutes-long audio fragments.

Two main factors contribute to the choice of base in the exponentiation of $w_k$ and $w_{k,x}$, $w_{k,y}$ frequencies: i) how precise the location should be for encoding (e.g. there is no need to encode position in audio much more precisely than one syllable takes); ii) longest reasonable "length" that is expected in the data. CAPE could be parametrized by precision and "length" parameters, but we believe that for all practical cases provided choices are reasonable.

---

[2]Work [50] introduces general complex-valued embeddings as continuous word functions over a position.
[3]Authors of [7] also perform coordinates scaling but it spans interval $[0, 1]$.

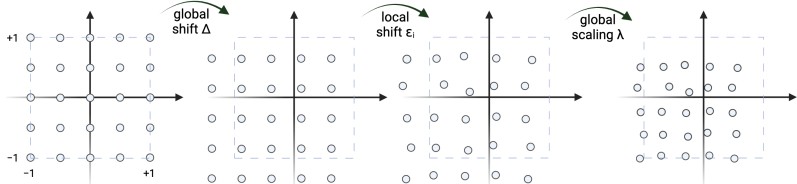

Figure 1: Example of CAPE's transformations for an image: patches positions are scaled to $[-1, 1] \times [-1, 1]$; random global shift, local shift, and global scaling are then applied to the grid of positions.

## 4  Continuous Augmented Positional Embeddings (CAPE)

Absolute sinusoidal positional embeddings lack regularization, leading to "in-domain" generalization issues as the model may start learning spurious correlations. For applications where inference input sizes may be significantly different than the training ones, "out-of-domain" issues may also arise, as positions rarely observed at training may lead to improper inference predictions. To ensure that the model does not learn spurious correlations between content and position, we introduce the following augmentations for absolute sinusoidal positional embeddings at training time, see Figure 1.

**Global shift**   Transform every embedding in a sequence using $S^{\Delta}$ operator with a global random shift from uniform zero-mean distribution $\Delta \sim \mathcal{U}(-\Delta_{max}, \Delta_{max})$:

$$\mathbf{E}'(n) = S^{\Delta} \mathbf{E}(n) \qquad \{S^{\Delta} \mathbf{X}\}_k = X_k e^{i\omega_k \Delta} \qquad \Delta \in \mathbb{R}$$

This modification hides the absolute positional information, but relative relations, see Eq. (2), between embeddings still hold. This transformation can be rewritten as augmenting positions by a random shift before encoding with $\sin$ and $\cos$:

$$n'_i \leftarrow n_i + \Delta \qquad x'_i \leftarrow x_i + \Delta_x, \ y'_i \leftarrow y_i + \Delta_y \qquad t'_i \leftarrow t_i + \Delta \qquad (5)$$

**Local shift**   To further increase augmentation and prevent capturing spontaneous correlations, we additionally introduce local shifts from uniform zero-mean distribution $\epsilon_i \sim \mathcal{U}(-\epsilon_{max}, \epsilon_{max})$

$$n'_i \leftarrow n_i + \epsilon_i \qquad x'_i \leftarrow x_i + \epsilon_{x,i}, \ y'_i \leftarrow y_i + \epsilon_{y,i} \qquad t'_i \leftarrow t_i + \epsilon_i \qquad (6)$$

**Global scaling**   To prevent distances memorization, we also introduce random global scale $\lambda$ from $\log \lambda \sim \mathcal{U}(-\log \lambda_{max}, \log \lambda_{max})$

$$n'_i \leftarrow \lambda n_i \qquad x'_i \leftarrow \lambda x_i, \qquad y'_i \leftarrow \lambda y_i \qquad t'_i \leftarrow \lambda t_i \qquad (7)$$

At training time, computing our continuous augmented positional embedding is performed through four steps: i) mean-normalization of positions (extract mean of sequence positions), ii) global shift Eq. (5), iii) local shift Eq. (6), and iv) global scaling Eq. (7). At inference time, only mean-normalization of positions is performed. The reference implementation can be found in Appendix A.

## 5  Experiments

### 5.1  Image Recognition

We evaluate CAPE embedding empirically with a recently proposed Vision Transformer (ViT) [16, 47] for image recognition. These works rely on learnable absolute positional embedding (*abspos*) for both class token and patches, and train ViT models on $224^2$ images[4] with $16^2$ patches. To further improve model quality, [47] performs fine-tuning on images of higher resolution $384^2$. The grid of positional embeddings is then upsampled.

---

[4]In the following, we denote the size and resolution $N \times N$ as $N^2$.

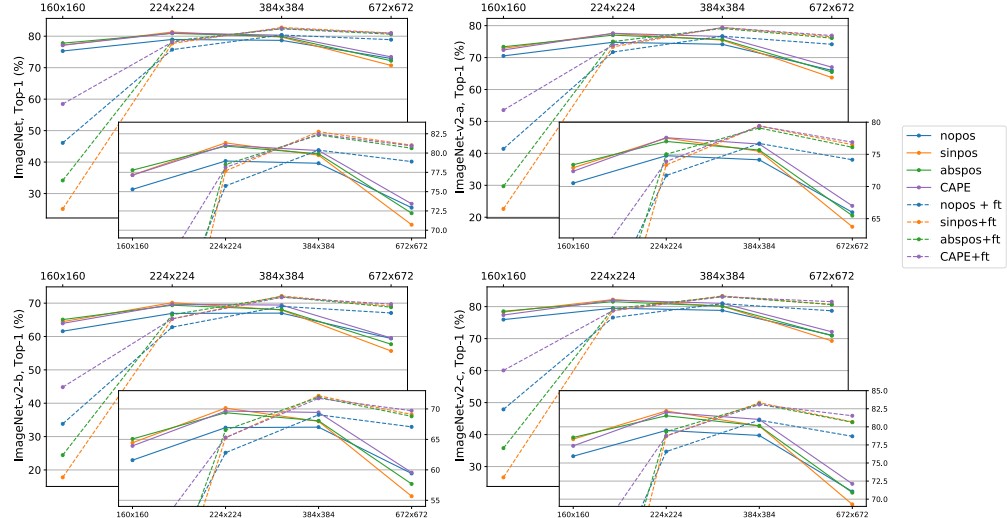

Figure 2: Top-1 accuracy on ImageNet and ImageNet-v2 for ViT models trained with different positional embeddings on $224^2$ resolution (solid) and further fine-tuned on $384^2$ (dashed, "+ft"). Insets focus on higher accuracies. The full list of top-1 and top-5 accuracies can be found in Appendix B.1, Tables 3 and 4.

**Data and ViT Models** All experiments are performed on the ImageNet [13, 43] dataset. We report top-1 and top-5 accuracies on ImageNet validation set and ImageNet-v2-{a,b,c} [40] test sets. The same convolution-free architecture, identical to the one proposed by [16] (ViT-B) and used by [47] (referred as DeiT-B), is chosen for all experiments. A ViT-B/DeiT-B baseline is trained with *abspos* on $224^2$ images, carefully following Section 6 from [47]. The *exact same training configuration* is used for models with other positional embeddings: *only* positional embedding is changed. We evaluate both proposed absolute sinusoidal positional embedding (*sinpos*), Eq. (3-4), and CAPE ($\Delta_{max} = 0.5$, $\epsilon_{max} = 1/N$ and $\lambda_{max} = 1.4$). As a control experiment we also train a model without any positional embedding (*nopos*), which can be interpreted as a 'bag of ~~words~~ patches', as no patch position information is available. We also train models with different positional embeddings on either $160^2$ or $384^2$ images. The whole training configuration remains the same as for training on $224^2$ images, except for the positional embedding. All models trained on $224^2$ images we additionally fine-tune on images of higher resolution $384^2$, following [47].

**Evaluation** To study generalization when image sizes vary, we evaluate all models on different resolutions ($160^2$, $224^2$, $384^2$ and $672^2$) by resizing all images in validation and test sets. When evaluating on resolutions different from the training one, bicubic interpolation is applied to *abspos* embeddings[5] as was justified in [47]. In contrast, *sinpos* and CAPE approaches can ingest any image resolution, thanks to the continuous nature of their positional embeddings.

### 5.1.1 Results

In Figure 2 we compare generalization performance of models trained with different positional embeddings on $224^2$ images (solid). Both proposed *sinpos* and CAPE approaches perform at least as well, if not better, than the *abspos* approach on the same-as-train resolution. When performing inference on resolutions different than the training one, CAPE performs best, notably outperforming *abspos* on high ($672^2$) and low ($160^2$) resolutions up to 25% and 2%, respectively. On $160^2$ and $384^2$ resolutions CAPE trained on $224^2$ resolution performs similar to *abspos* trained on corresponding $160^2$ or $384^2$ inference resolutions (the latter results being reported in Figure 3). This confirms good generalization properties of CAPE on image resolutions unseen at training time.

*Abspos* fine-tuned on a higher resolution ($384^2$) improves in accuracy for both $384^2$ and $672^2$ resolutions, while degrading on lower ones (original $224^2$ and lowest $160^2$), as shown in Figure 2

---

[5]Interpolation is not applied to the class token embedding.

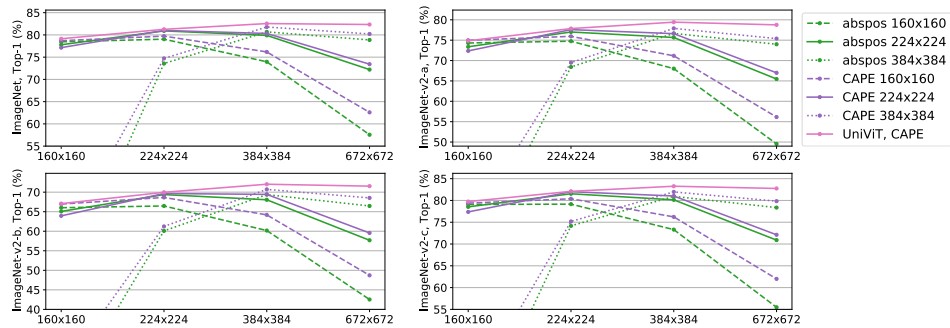

Figure 3: Top-1 accuracy on ImageNet and ImageNet-v2 for ViT models with either *abspos* or CAPE trained on each particular resolution and UniViT model trained on the mixture of resolutions.

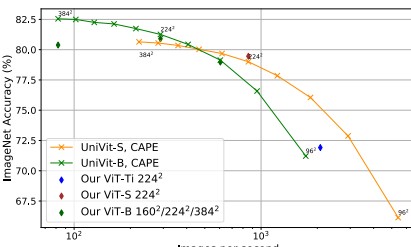

Figure 4: Accuracy with respect to throughput on ImageNet at inference time under variation of image resolution. "-S" and "-Ti" refer to small and tiny architectures [16], respectively.

Table 1: Test time augmentation (TTA) results on ImageNet when predictions on resolutions $r^2$, $(r+32)^2$ and $(r-32)^2$ are combined. *Sinpos* and CAPE are trained on $224^2$ resolution.

| Model | $r$ | Top-1 (%) | | Top-5 (%) | |
|---|---|---|---|---|---|
| | | -TTA | +TTA | -TTA | + TTA |
| sinpos | 224 | 81.32 | 81.47 | 95.44 | 95.54 |
| CAPE | 224 | 81.01 | 81.34 | 95.18 | 95.49 |
| UniViT, sinpos | 224 | 80.82 | 81.34 | 95.40 | 95.57 |
| UniViT, CAPE | 224 | 81.26 | 81.64 | 95.56 | 95.71 |
| UniViT, sinpos | 384 | 82.31 | 82.44 | 96.04 | 96.14 |
| UniViT, CAPE | 384 | 82.55 | 82.72 | 96.18 | 96.22 |

(dashed). *Sinpos* and CAPE fine-tuned on $384^2$ resolution outperform *abspos* by 0.3-0.4%, thanks to a better fine-tuning starting point after resolution change. In that setting, *sinpos* and CAPE keep better generalization performance on nearby resolutions ($224^2$ and $672^2$). While being seriously impacted on the $160^2$ resolution, CAPE still outperforms others by 12-25%. Comparison of ViT models trained with either *abspos* or CAPE on each particular resolution is shown in Figure 3. CAPE outperforms *abspos* on a specific training resolution, and generalizes better to unseen resolutions.

**No Positional Embedding** With *nopos* model, we confirm [16]'s observation that positional embedding does not have a critical importance for ImageNet classification (see Figure 2). *Nopos* model's simplicity and generalization abilities make it a nice baseline for positional embedding study in computer vision. It has generalization accuracy similar to *abspos* on low $160^2$ and high $672^2$ resolutions, while CAPE outperforms *nopos* across the board. It is likely that *abspos* suffers from the embedding interpolation step on extreme resolutions. In contrast, both *sinpos* and CAPE have the advantage to naturally support variation in resolutions.

### 5.1.2 UniViT: Training Universal Transformer on Different Resolutions

As CAPE-based models can handle any image resolution, we propose a single universal ViT model, called UniViT, which is trained on images of different resolutions: during training we resize all images in a batch to a randomly chosen size, uniformly sampled in the range $[128, 320]^2$ with a step of 32. For experiments with UniViT training configuration remains the same as for ViT. Because of training on the images of different resolution UniViT training time is only 1.1x longer than ViT trained on $224^2$ resolution. In Figure 3 we compare UniViT model trained with CAPE ($\lambda = 1$) against different ViT models trained on each particular resolution: UniViT outperforms single-resolution ViT models for any given resolution and, moreover, generalizes to non-training resolutions well.

Image resolution directly impacts throughput: computational complexity of attention is $O(N^4)$ for a $N \times N$ image. In Figure 4, we show UniViT with CAPE throughput and accuracy with respect to input image resolution. On $96^2$ resolution UniViT with CAPE handles throughput and accuracy

similar to "tiny" vanilla ViT, while "small" UniViT with CAPE has 4% higher accuracy (with identical throughput) on resolution $160^2$ and 1.4x higher throughput (with identical accuracy) on resolution $128^2$. Thus, UniViT unlocks the possibility of dynamically adjusting throughput of a model in a production regime under heavy loads, a practical alternative to improving model throughput at inference time via decreasing model size.

We further improve UniViT with CAPE accuracy by resizing each image to its optimal resolution at inference, as shown in Appendix B.2 Table 5. We split ImageNet validation images into 8 bins, according to their size. By selecting an optimal resizing strategy in each bin we are able to improve top-1 accuracy to 82.92% (in comparison, the model has 81.26% on $224^2$ and 82.55% on $384^2$).

### 5.1.3 Resizing as Test Time Augmentation (TTA)

As both *sinpos*- and CAPE-based models handle well different image resolutions, we propose to perform test time resolution augmentation when evaluating a single model. For TTA we average model's logits evaluated on three resolutions for the same image: $r^2$, $(r-32)^2$ and $(r+32)^2$, where $r$ is either 224 or 384. As shown in Table 1, ViT and UniViT models trained with either *sinpos* or CAPE embeddings get 0.2%-0.5% top-1 accuracy boost with this test time augmentation.

### 5.2 Automatic Speech Recognition (ASR)

Recently it was shown that Transformer [48] architectures are state-of-the-art on different public benchmarks for ASR [55, 33, 30, 8].

**Data** We evaluate our models on several English speech datasets, both on in-domain data and out-of-domain data. We also analyze our models generalization to long sequences. We consider two standard training benchmarks: Wall Street Journal (WSJ) [20, 29, 52], read speech with 81.5h of training data, and TED-LIUM v3 (TL) [24], oratory speech with 452h of training data. Besides these datasets we use two other sets for evaluation only: i) LibriSpeech (LS) [36], read speech from audiobook recordings (we use only test sets with clean, *test-clean*, and noisy, *test-other*, speech); ii) Robust Video (RV), our in-house English video dataset, which is sampled from public social media videos and aggregated and de-identified before transcription; these videos contain a diverse range of speakers, accents, topics, and acoustic conditions making ASR difficult; the test sets are composed of *clean* and *noisy* subsets. Details on data and its statistics can be found in Appendix C.1.

**Evaluation** To evaluate our acoustic models on sequence lengths not seen at training time, we split all evaluation utterances by their duration $T$ into the following bins: $T < 10$s, $T \in [10-15)$s, $T \in [15, 20)$s and $T >= 20$s. Our performance metric is word error rate (WER) (no language model is involved), reported for each sequence length bin and for the entire evaluation dataset. For RV data a hand-crafted segmentation is available, allowing us to evaluate on the exact same data, but segmented in different ways. More precisely, for RV data we prepare 8 sets where audios have the following respective durations: $T = 10, 15, 20, 25, 30, 35, 40, 45$s.

**Acoustic Model (AM) Training** All models are trained with Connectionist Temporal Classification [22]. SpecAugment [37] is used as data augmentation in training, and the network architecture follows [30]: the AM encoder is composed of a 1D convolution (kernel 7, stride 3) with a GLU activation and 36 4-heads Transformer layers [48], finally followed by a linear layer which outputs a score for each target token. Our token set consists of 26 English alphabet letters, augmented with the apostrophe and a word boundary token (further details in Appendix C.2).

**Positional Embedding** As in vision experiments, we evaluate *nopos*, *sinpos*, *abspos* and CAPE-based models. In addition, we evaluate models trained with relative positional embeddings: in that case, no absolute positions are used, and learnable relative positional embeddings [44] (*relpos*) are trained in each Transformer layer. We follow [30] to train an AM baseline with *relpos*. For models with other positional embeddings, the training *configuration remains identical*. Abspos $\{\mathbf{E}(t)\}_{t=1}^{N}$ is set to cover 13.8s of context. At training/evaluation time for the longer sequences we define *abspos* as $\mathbf{E}(t) \equiv \mathbf{E}(t \mod N)$ for $t > N$. This extrapolation at training time leaves a chance to the acoustic model to generalize to unseen (longer) sequence lengths. *Relpos* spans a large context, 26.8s to the left/right. CAPE's global shift covers 60s, while a local shift is set to its maximum to preserve the frames order; $\lambda_{max} = 1.1$ and $\lambda_{max} = 2$ for WSJ and TL, respectively.

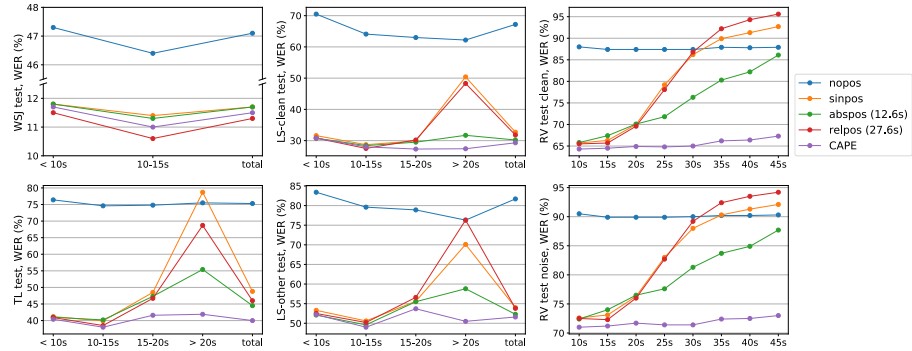

Figure 5: Word error rate for models trained on WSJ with different positional embeddings.

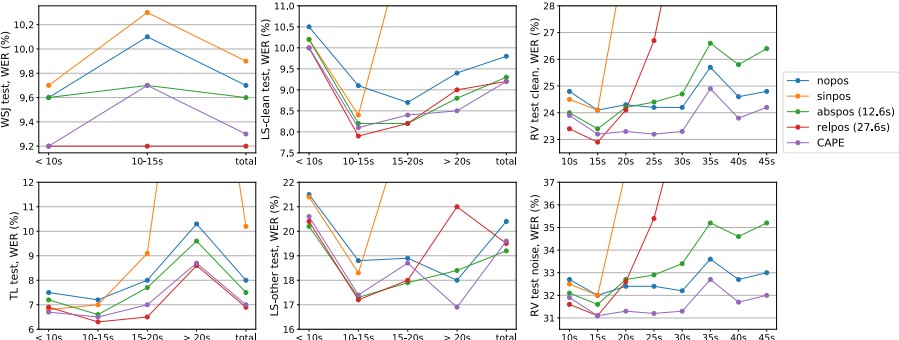

Figure 6: Word error rate for models trained on TED-LIUM v3 with different positional embeddings.

### 5.2.1 Results

A model trained on WSJ with CAPE outperforms other positional embeddings on both public and RV data across different audio durations, as shown in Figure 5. A model trained on TL with CAPE outperforms *nopos* and *sinpos* on all data, outperforms *abspos* and *relpos* for audio longer than 20s, and behaves similarly on shorter durations, see Figure 6. On RV data, CAPE-based models perform uniformly well on different audio durations, including long ones. In contrast, other embeddings-based models are seriously impacted when audio duration increases. Finally, CAPE does not have computational or parameters overheads compared to *relpos*.

**No Positional Embedding** As expected, *nopos* models (both WSJ and TL ones) perform similarly in WER across different audio durations. However, *nopos* TL model performs surprisingly well: it is competitive to positional embeddings-based models on public data. On RV data, *nopos* TL model outperforms all other models, except CAPE when $T > 20$s. We perform ablations in Appendix C.4 to show that key ingredients are CTC loss, sufficient model capacity, and large amount of data for this effect to occur.

**CAPE as Augmentation** CAPE can be viewed as a data augmentation performed on input data (positions), which regularizes the trained model. We demonstrate this by training on TL with either *sinpos* or CAPE and with/without SpecAugment (no other augmentations are used), see Figure 7. Baseline *sinpos* without any augmentation performs the worst with a large gap. Including either CAPE or SpecAugment decreases the WER significantly by 4-5%: SpecAugment is more beneficial due to its domain relevance. Combining together CAPE and SpecAugment further decreases the WER by 2.5-3.5%, showing that augmentations are complementary to each other.

### 5.2.2 Padding-free ASR with CAPE and Variable STFT Hop Distance

In ASR, when batching audios of different duration, one often relies on padding tokens. We propose instead to perform time stretching augmentation on all audios in the batch, such that they will have the

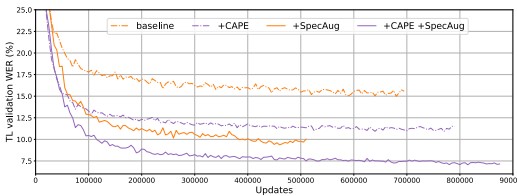

Figure 7: Validation WER for models trained with different augmentations: "baseline" is a model with *sinpos*, "+CAPE" adds CAPE's global and local shifts, "+SpecAug" adds SpecAugment.

Table 2: WMT'14 BLEU score (3 runs avg.).

| Model & Lang. | Embedding | Valid BLEU | Test BLEU |
|---|---|---|---|
| 6L-6L | sinpos | 26.88±0.05 | 27.66±0.10 |
| | abspos | 26.68±0.05 | 27.36±0.06 |
| | relpos | 26.81±0.16 | 27.92±0.07 |
| DE | CAPE, $\Delta = 5$ | 26.86±0.13 | 27.89±0.07 |
| | CAPE, $\Delta = 50$ | 27.09±0.03 | 27.77±0.16 |
| 18L-18L | sinpos | 27.09±0.06 | 28.28±0.28 |
| | abspos | 27.23±0.02 | 28.26±0.22 |
| DE | CAPE, $\Delta = 10$ | 27.17±0.10 | 28.44±0.06 |
| 6L-6L | sinpos | 47.27±0.03 | 41.13±0.07 |
| | abspos | 47.22±0.03 | 41.21±0.04 |
| | relpos | 47.12±0.03 | 41.33±0.13 |
| FR | CAPE, $\Delta = 5$ | 47.22±0.03 | 41.59±0.03 |
| | CAPE, $\Delta = 50$ | 47.14±0.02 | 41.48±0.10 |

same number of frames. We perform this augmentation by tuning the short-time Fourier Transform (STFT) hop distance when computing the audio features. Positions remain tied to the original audio timestamps. These models trained either on WSJ or TL show better WER across the board, compared to models trained with a padding approach, as shown in Appendix C.3 Figures 15 and 16, and improvements on RV are quite consistent. As before, CAPE-based models outperform *sinpos* models.

We found this padding-free approach convenient, as it *alleviates the implementation of special cases to handle padding tokens* in ASR models: e.g. any normalization should be aware which tokens are padding, otherwise normalization constants would depend on the amount of padding; any attention module should be aware which tokens it should not attend to. Moreover, for efficient computations and reducing padding tokens audio samples are often packed together via sorting by their duration; this reduces variability in the batches drastically. Our results demonstrate that with CAPE and padding-free approach we can mix samples of not-too-different lengths within a batch, providing better randomization and utilize all frames. While UniViT adjusts number of tokens by changing resolution, the STFT hop distance achieves the same for audio. By adjusting the hop distance during inference for padding-free ASR, we can achieve higher throughput with similar recognition quality, see Appendix C.3 Figure 17. However, very low resolution (e.g. 128x128) in images works well enough while speech recognition is far more sensitive to token sparsification as phoneme can last as little as 30-50ms.

## 5.3 Machine Translation (MT)

Our MT experiments follow the recent results with Transformers combined with a new initialization scheme (ADMIN) [31, 32]. This approach allows to train very deep state-of-the-art Transformers for MT. We did not implement back-translation or other domain-specific augmentations.

**Data and Models Training** Experiments are conducted on standard WMT'14 English-French (FR) and English-German (DE) benchmarks. For both benchmarks we follow [31, 32]: for FR we use a 40k subword vocabulary, and evaluate on the provided "valid" file for validation and newstest14 for test. On DE, we consider a 32K subword vocabulary, newstest2013 for validation, and newstest2014 for test. We reproduce results from [31, 32] by training a *sinpos*-based model with 6L-6L, 18L-18L for DE and 6L-6L for FR encoder-decoder layers with ADMIN. *Training configuration stays the same* for other positional embedding-based models, other than positional embeddings being either *abspos* (covers 1k tokens), *relpos* (learnable [44], covers max train content: 150 left/right tokens) or CAPE in both encoder and decoder layers. For CAPE to have approximate correspondence in positions of source and target sentences, we first scale positions of source language by a factor $\alpha = \frac{\text{\# tokens in target corpus}}{\text{\# tokens in source corpus}} \in \mathbb{R}$ so that positions interval is loosely matched between source and target sentences. For each training sample we then apply the same global shift and scale for source and target positions. Local shifts for source/target positions are independently sampled from $\mathcal{U}(-0.5, 0.5)$. As no absolute positions are provided anymore, we prepend source sentences with a "begin of sentence" token to hint the decoder with a first position during both training and evaluation.

**Evaluation** We select the best checkpoint according to BLEU on the validation set, using a beam size 4 for DE and 5 for FR. Following convention, BLEU is computed by `multi-bleu.perl` via the

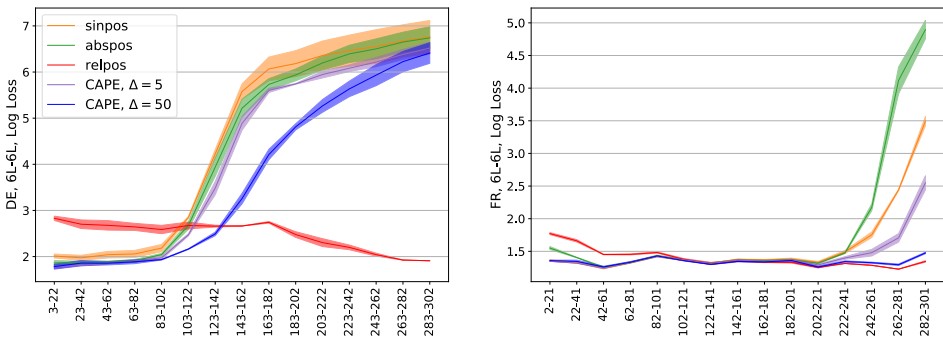

Figure 8: Average negative log loss across consecutive positions measured on test sets with stacked sentences for models with different positional embeddings: 6L-6L DE (left) and 6L-6L FR (right).

standardized tokenization of the publicly-accessible dataset. As WMT'14 test sets have limited length of sentences being same as train we artificially stack together sentences in test sets (source and target) to have 300+ tokens in each new target sample (3k original pairs of sentences are transformed into 270 pairs): during stacking dots on junctions are replaced with commas and first letters of the next stacked sentence are lower-cased to match one-sentence setup. On these test sets we compute log loss for each position: lower value shows that a model is more likely to predict a correct word at a particular position.

**Results**   Comparison between models trained with different positional embeddings on WMT'14 benchmarks is shown in Table 2. CAPE outperforms *sinpos* and *abspos* on all settings, is similar to *relpos* on DE and outperforms it on FR. Figure 8 shows that i) for positions covered in training (<150) all absolute positional embeddings behave similarly and outperform *relpos*; ii) for positions not seen during training (>200) *relpos* outperforms others. However, CAPE generalizes better than *abspos* and *sinpos* and, moreover, is able to generalize similar to *relpos* with more data (FR).

# 6   Discussion and Conclusion

Encoding positional information is a key component of attention-based models. Poor generalization of absolute sinusoidal positional embeddings led to numerous works investigating ways to encode relevant positional information. Existing solutions are often modality-specific, non-trivial to implement and incur computational overhead. We demonstrated in different domains that existing positional embeddings may generalize poorly in certain conditions. We introduced a simple and efficient continuous augmented positional embedding, CAPE, which preserves some information about relative token positions. Thanks to its continuous nature, CAPE allows augmentations which were previously not possible. CAPE makes models more flexible both at training and inference. It generalizes well to input sizes not seen during training across a variety of domains. We expect emergence of new training and production pipelines that leverage the adjustable throughput property when tuning the input size: CAPE provides a relatively simple way of producing more efficient Transformer by down-sampling input (e.g for image and audio). Going further, CAPE-based architectures are free from baked-in restrictions on patches positions: these could overlap, or be sparse for example – opportunities impossible for convolution-containing Transformers. In contrast to relative positional embeddings which modify attention mechanism, CAPE is ready to be used by novel attention mechanisms, such as "linear" Transformers [46]. Finally, CAPE can be combined with relative positional embeddings like [11] and [45] to limit over-fitting to exact relative positions.

**Limitations**   CAPE applies only to attention-based models, and no testing was performed outside described modalities. From a representation perspective we demonstrated that CAPE is capable of providing relative positioning, however, a model should "learn" it. Thus, we can expect that relative positional embeddings should be beneficial in settings with small amount of data because of more appropriate inductive bias. Proposed UniViT model was tested only for image recognition task and further exploration of broader UniViT applicability to other tasks is a subject of future work.

# 7 Acknowledgments

We would like to thank Mark Tygert and Edouard Grave for relevant references and helpful discussions, Paden Tomasello for English language editing.

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
