# A    CAPE Implementation in Python

```python
import numpy as np

def augment_positions_1d(
    positions_1d: np.ndarray,
    mean_normalize: bool,
    augment: bool,     # True during training
    max_global_shift, # delta max
    max_local_shift,  # epsilon max
    max_scale,        # lambda max
    rng=np.random.RandomState(42)
):
    """
    Takes original positions, returns modified ones.
    Can reuse sin/cos embedding from "Attention is all you need".
    Code handles NaNs is positions_1d input as if those correspond to pad tokens
    """
    assert max_scale >= 1
    batch_size, n_tokens = positions_1d.shape
    if mean_normalize:
        positions_1d -= np.nanmean(positions_1d, axis=1, keepdims=True)
    if augment:
        delta = rng.uniform(-max_global_shift, +max_global_shift, size=[batch_size, 1])
        delta_local = rng.uniform(-max_local_shift, +max_local_shift, size=[batch_size, n_tokens])
        log_lambdas = rng.uniform(-np.log(max_scale), +np.log(max_scale), size=[batch_size, 1])
        new_positions = (positions_1d + delta + delta_local) * np.exp(log_lambdas)
        return new_positions
    else:
        return positions_1d

def CAPE_2d(
    n_patches: int,    # number of patches, default in ViT is 14
    batch_size: int,
    augment: bool,     # True during training
    n_channels: int,   # embedding size for one patch
    max_global_shift, # delta max
    max_local_shift,  # epsilon max
    max_scale,        # lambda max
    rng=np.random.RandomState(42)
):
    """Prepares grid of CAPE embeddings for provided grid size"""
    x = np.zeros([batch_size, n_patches, n_patches])
    y = np.zeros([batch_size, n_patches, n_patches])
    x += np.linspace(-1, 1, n_patches)[None, :, None]
    y += np.linspace(-1, 1, n_patches)[None, None, :]
    if augment:
        # global shift
        x += rng.uniform(-max_global_shift, +max_global_shift, size=[batch_size, 1, 1])
        y += rng.uniform(-max_global_shift, +max_global_shift, size=[batch_size, 1, 1])
        # local shift
        x += rng.uniform(-max_local_shift, +max_local_shift, size=x.shape)
        y += rng.uniform(-max_local_shift, +max_local_shift, size=y.shape)
        # scaling
        lambdas = np.exp(rng.uniform(-np.log(max_scale), + np.log(max_scale), size=[batch_size, 1, 1]))
        x *= lambdas
        y *= lambdas

    assert n_channels % 2 == 0
    half_channels = n_channels // 2
    rho = 10 ** (np.arange(1, half_channels + 1) / half_channels)
    # recommended simpler approximate implementation
    # rho = 10 ** np.linspace(0, 1, half_channels)
    w_x = rho * np.cos(np.arange(half_channels))
    w_y = rho * np.sin(np.arange(half_channels))

    phase = np.pi * (w_x * x[:, :, :, None] + w_y * y[:, :, :, None])
    return np.concatenate([np.cos(phase), np.sin(phase)], axis=-1)
```

## B  Image Recognition Experiments

### B.1  Technical Details

Table 3: Top-1 accuracy (%) for ViT and UniViT models evaluated on ImageNet validation set and ImageNet-v2 test sets with images resized to different resolutions: $160^2$, $224^2$, $384^2$ and $672^2$. Models trained on $224^2$ and further fine-tuned on $384^2$ resolution are marked with "+ft". "-S" and "-Ti" refer to small and tiny architectures [16], respectively.

| Model | Train Res. | ImageNet | | | | ImageNet-v2-a | | | | ImageNet-v2-b | | | | ImageNet-v2-c | | | |
|---|---|---|---|---|---|---|---|---|---|---|---|---|---|---|---|---|---|
| | | $160^2$ | $224^2$ | $384^2$ | $672^2$ | $160^2$ | $224^2$ | $384^2$ | $672^2$ | $160^2$ | $224^2$ | $384^2$ | $672^2$ | $160^2$ | $224^2$ | $384^2$ | $672^2$ |
| nopos | $160^2$ | 77.33 | 78.38 | 75.46 | 61.00 | 72.39 | 73.78 | 69.65 | 52.99 | 64.41 | 66.25 | 62.58 | 46.58 | 77.67 | 78.76 | 75.28 | 59.13 |
| abspos | | 78.45 | 79.04 | 73.96 | 57.56 | 74.31 | 74.75 | 68.02 | 49.54 | 66.02 | 66.47 | 60.18 | 42.53 | 79.08 | 79.15 | 73.31 | 55.48 |
| sinpos | | 79.05 | 74.38 | 65.65 | 44.13 | 74.75 | 70.11 | 59.61 | 36.75 | 66.91 | 62.13 | 52.28 | 30.55 | 79.37 | 75.41 | 65.45 | 42.51 |
| CAPE, $\lambda = 1$ | | 78.74 | 79.69 | 75.82 | 61.87 | 74.94 | 75.75 | 70.50 | 54.91 | 66.56 | 68.17 | 62.78 | 46.74 | 79.51 | 80.18 | 75.71 | 60.39 |
| CAPE | | 78.70 | 79.73 | 76.18 | 62.59 | 74.98 | 75.94 | 71.17 | 56.13 | 66.95 | 68.65 | 64.19 | 48.72 | 79.38 | 80.38 | 76.23 | 62.00 |
| nopos | $224^2$ | 75.30 | 78.97 | 78.68 | 72.92 | 70.52 | 74.79 | 74.15 | 65.99 | 61.59 | 66.93 | 67.01 | 59.41 | 75.94 | 79.50 | 78.81 | 71.06 |
| abspos | | 77.78 | 80.90 | 79.90 | 72.21 | 73.38 | 77.01 | 75.66 | 65.47 | 65.07 | 69.38 | 68.05 | 57.69 | 78.53 | 81.53 | 80.14 | 70.90 |
| sinpos | | 77.15 | 81.32 | 79.72 | 70.71 | 72.91 | 77.52 | 75.48 | 63.74 | 64.45 | 70.14 | 67.96 | 55.67 | 78.30 | 82.19 | 80.13 | 69.28 |
| CAPE, $\lambda = 1, \Delta = 0$ | | 77.53 | 81.08 | 80.18 | 72.34 | 73.41 | 77.61 | 75.81 | 65.99 | 64.85 | 70.23 | 68.22 | 57.83 | 78.33 | 82.05 | 80.06 | 70.99 |
| CAPE, $\lambda = 1, \epsilon = 0$ | | 77.46 | 81.14 | 80.49 | 72.13 | 73.35 | 77.90 | 76.36 | 64.95 | 64.84 | 69.84 | 68.45 | 57.31 | 78.80 | 81.84 | 80.58 | 70.53 |
| CAPE, $\lambda = 1$ | | 77.70 | 81.01 | 80.38 | 73.06 | 72.88 | 77.67 | 76.23 | 67.35 | 64.79 | 70.25 | 69.36 | 59.59 | 78.16 | 82.22 | 80.96 | 72.07 |
| CAPE, $\Delta = 0$ | | 77.35 | 81.08 | 80.50 | 73.11 | 73.05 | 77.59 | 76.73 | 67.25 | 64.98 | 69.75 | 69.14 | 59.01 | 78.32 | 81.88 | 81.00 | 72.04 |
| CAPE, $\epsilon = 0$ | | 77.71 | 81.30 | 80.57 | 73.35 | 73.51 | 77.71 | 76.72 | 66.58 | 65.01 | 69.93 | 69.59 | 59.31 | 78.02 | 81.93 | 80.86 | 71.73 |
| CAPE | | 77.14 | 81.01 | 80.33 | 73.43 | 72.37 | 77.59 | 76.60 | 66.99 | 63.94 | 69.65 | 69.43 | 59.56 | 77.37 | 82.00 | 81.02 | 72.12 |
| abspos | $384^2$ | 21.83 | 73.57 | 80.68 | 78.87 | 19.99 | 68.43 | 76.66 | 74.00 | 16.23 | 60.09 | 69.37 | 66.47 | 24.02 | 74.17 | 80.94 | 78.36 |
| sinpos | | 7.71 | 75.55 | 82.42 | 80.73 | 7.16 | 70.87 | 78.82 | 76.21 | 5.26 | 62.29 | 71.32 | 68.82 | 8.72 | 75.57 | 82.82 | 80.47 |
| CAPE, $\lambda = 1$ | | 31.98 | 75.38 | 82.55 | 80.94 | 28.47 | 71.00 | 78.97 | 75.85 | 23.51 | 62.40 | 71.55 | 69.02 | 34.15 | 76.25 | 82.77 | 80.60 |
| CAPE | | 32.97 | 74.71 | 81.78 | 80.22 | 29.23 | 69.50 | 77.90 | 75.39 | 24.21 | 61.23 | 70.71 | 68.53 | 34.83 | 75.18 | 81.96 | 79.83 |
| nopos+ft | $224^2 \downarrow 384^2$ | 46.10 | 75.74 | 80.38 | 78.89 | 41.42 | 71.72 | 76.69 | 74.16 | 33.82 | 62.81 | 69.04 | 67.06 | 47.87 | 76.58 | 80.92 | 78.69 |
| abspos+ft | | 34.16 | 78.51 | 82.35 | 80.63 | 29.71 | 75.02 | 79.10 | 76.10 | 24.45 | 66.54 | 71.91 | 68.79 | 35.76 | 79.33 | 83.15 | 80.63 |
| sinpos+ft | | 25.12 | 77.69 | 82.77 | 81.02 | 22.61 | 73.35 | 79.42 | 76.50 | 17.77 | 65.27 | 72.16 | 69.10 | 26.62 | 78.72 | 83.33 | 80.68 |
| CAPE+ft, $\lambda = 1$ | | 57.80 | 78.63 | 82.67 | 81.28 | 53.25 | 74.87 | 79.19 | 77.30 | 44.91 | 67.07 | 72.21 | 70.20 | 59.69 | 79.72 | 83.60 | 81.59 |
| CAPE+ft | | 58.46 | 78.14 | 82.46 | 81.53 | 53.55 | 73.89 | 79.40 | 76.89 | 44.82 | 65.51 | 71.74 | 69.71 | 60.05 | 78.76 | 83.08 | 81.53 |
| UniViT, sinpos | mix | 78.94 | 80.82 | 82.31 | 82.12 | 74.64 | 77.21 | 78.58 | 78.29 | 66.57 | 69.74 | 71.48 | 71.42 | 79.53 | 81.67 | 82.82 | 82.53 |
| UniViT, CAPE $\lambda = 1$ | | 79.14 | 81.26 | 82.55 | 82.34 | 74.85 | 77.85 | 79.42 | 78.76 | 67.07 | 69.97 | 72.03 | 71.54 | 79.74 | 82.09 | 83.26 | 82.75 |
| UniViT, CAPE | | 79.05 | 81.16 | 82.28 | 81.83 | 74.88 | 77.50 | 78.96 | 77.87 | 67.08 | 69.88 | 72.01 | 70.99 | 79.78 | 82.06 | 83.10 | 82.21 |
| abspos-S | $224^2$ | 74.89 | 79.46 | 77.83 | 64.32 | 70.89 | 75.83 | 73.45 | 57.40 | 62.04 | 68.12 | 65.86 | 49.22 | 76.15 | 80.54 | 78.52 | 63.24 |
| UniViT-S, CAPE, $\lambda = 1$ | mix | 76.05 | 79.00 | 80.64 | 80.31 | 72.57 | 75.56 | 77.59 | 76.66 | 64.00 | 67.31 | 70.25 | 69.41 | 78.13 | 80.44 | 81.98 | 81.30 |
| abspos-Ti | $224^2$ | 64.76 | 71.91 | 70.21 | 56.12 | 61.03 | 68.78 | 66.30 | 49.62 | 51.88 | 59.67 | 58.09 | 42.71 | 67.97 | 74.13 | 71.92 | 55.63 |
| UniViT-Ti, CAPE, $\lambda = 1$ | mix | 65.25 | 69.83 | 72.44 | 71.15 | 62.20 | 66.40 | 68.99 | 67.45 | 53.35 | 57.25 | 61.30 | 59.72 | 69.15 | 72.64 | 74.93 | 73.23 |

Table 4: Top-5 accuracy (%) for ViT and UniViT models evaluated on ImageNet validation set and ImageNet-v2 test sets with images resized to different resolutions: $160^2$, $224^2$, $384^2$ and $672^2$. Models trained on $224^2$ and further fine-tuned on $384^2$ resolution are marked with "+ft". "-S" and "-Ti" refer to small and tiny architectures [16], respectively.

| Model | Train Res. | ImageNet | | | | ImageNet-v2-a | | | | ImageNet-v2-b | | | | ImageNet-v2-c | | | |
|---|---|---|---|---|---|---|---|---|---|---|---|---|---|---|---|---|---|
| | | $160^2$ | $224^2$ | $384^2$ | $672^2$ | $160^2$ | $224^2$ | $384^2$ | $672^2$ | $160^2$ | $224^2$ | $384^2$ | $672^2$ | $160^2$ | $224^2$ | $384^2$ | $672^2$ |
| nopos | $160^2$ | 93.20 | 93.99 | 92.43 | 83.69 | 91.71 | 92.64 | 90.22 | 77.75 | 84.88 | 86.39 | 83.96 | 71.52 | 94.60 | 95.02 | 93.09 | 82.41 |
| abspos | | 94.06 | 94.37 | 91.60 | 81.26 | 92.66 | 93.14 | 88.97 | 75.19 | 86.07 | 86.76 | 82.27 | 67.75 | 95.30 | 95.31 | 92.27 | 80.53 |
| sinpos | | 94.26 | 91.80 | 86.41 | 68.55 | 93.20 | 90.08 | 82.75 | 61.43 | 86.65 | 83.28 | 75.62 | 54.14 | 95.70 | 93.24 | 87.03 | 67.55 |
| CAPE, $\lambda = 1$ | | 94.19 | 94.78 | 92.92 | 84.57 | 92.84 | 93.46 | 90.55 | 79.44 | 86.87 | 87.75 | 84.62 | 72.17 | 95.44 | 95.80 | 93.85 | 84.43 |
| CAPE | | 94.15 | 94.80 | 93.03 | 85.32 | 92.95 | 93.65 | 90.82 | 80.48 | 86.86 | 87.86 | 85.22 | 73.47 | 95.27 | 95.72 | 94.06 | 84.92 |
| nopos | $224^2$ | 92.14 | 94.22 | 94.11 | 90.89 | 90.17 | 92.79 | 92.73 | 87.58 | 82.72 | 86.92 | 87.27 | 81.61 | 93.48 | 95.19 | 94.88 | 90.91 |
| abspos | | 93.45 | 95.26 | 94.71 | 90.49 | 91.80 | 93.91 | 93.24 | 87.13 | 85.34 | 88.52 | 87.77 | 80.23 | 94.52 | 96.17 | 95.43 | 90.76 |
| sinpos | | 93.29 | 95.44 | 94.52 | 89.77 | 91.48 | 94.22 | 92.84 | 85.33 | 84.52 | 88.75 | 87.22 | 78.37 | 94.48 | 96.46 | 95.23 | 89.03 |
| CAPE, $\lambda = 1, \Delta = 0$ | | 93.29 | 95.21 | 94.73 | 90.56 | 91.47 | 93.73 | 92.91 | 87.42 | 84.80 | 88.26 | 87.69 | 80.74 | 94.43 | 96.14 | 95.37 | 90.60 |
| CAPE, $\lambda = 1, \epsilon = 0$ | | 93.37 | 95.42 | 94.97 | 90.72 | 91.46 | 94.11 | 93.26 | 86.82 | 85.26 | 88.87 | 88.15 | 80.26 | 94.47 | 96.22 | 95.61 | 90.34 |
| CAPE, $\lambda = 1$ | | 93.45 | 95.45 | 95.09 | 91.50 | 91.69 | 94.11 | 93.64 | 87.84 | 85.61 | 88.87 | 88.52 | 81.79 | 94.63 | 96.18 | 95.91 | 91.60 |
| CAPE, $\Delta = 0$ | | 93.19 | 95.42 | 95.00 | 91.30 | 91.94 | 94.38 | 93.75 | 88.13 | 85.13 | 88.97 | 88.52 | 82.01 | 94.63 | 96.27 | 96.01 | 91.40 |
| CAPE, $\epsilon = 0$ | | 93.26 | 95.32 | 94.94 | 91.25 | 91.92 | 94.24 | 93.63 | 87.80 | 84.88 | 88.64 | 88.26 | 81.10 | 94.65 | 96.18 | 95.71 | 90.99 |
| CAPE | | 93.18 | 95.18 | 94.94 | 91.57 | 91.64 | 94.23 | 93.77 | 88.56 | 85.18 | 88.31 | 88.33 | 82.04 | 94.49 | 96.39 | 95.94 | 91.85 |
| abspos | $384^2$ | 38.45 | 90.69 | 94.99 | 93.89 | 35.76 | 87.99 | 93.39 | 91.93 | 30.10 | 80.81 | 88.09 | 86.19 | 40.91 | 91.60 | 95.58 | 94.49 |
| sinpos | | 15.86 | 91.88 | 95.68 | 94.79 | 14.71 | 89.88 | 94.30 | 93.11 | 11.98 | 82.13 | 89.54 | 87.71 | 17.18 | 92.82 | 96.29 | 95.23 |
| CAPE, $\lambda = 1$ | | 51.03 | 91.96 | 95.83 | 94.99 | 48.04 | 89.77 | 94.48 | 93.02 | 40.55 | 82.73 | 89.58 | 88.14 | 54.47 | 93.07 | 96.43 | 95.30 |
| CAPE | | 51.82 | 91.29 | 95.40 | 94.53 | 47.32 | 88.62 | 94.07 | 92.52 | 40.21 | 81.29 | 88.75 | 87.64 | 53.95 | 92.16 | 96.09 | 94.94 |
| nopos+ft | $224^2 \downarrow 384^2$ | 68.45 | 92.51 | 95.12 | 94.25 | 63.71 | 90.61 | 93.86 | 92.67 | 55.07 | 83.52 | 89.47 | 87.18 | 70.52 | 93.72 | 95.70 | 94.83 |
| abspos+ft | | 54.06 | 93.96 | 95.96 | 95.11 | 49.48 | 92.58 | 95.11 | 93.60 | 42.31 | 86.42 | 90.19 | 88.39 | 56.36 | 95.12 | 96.85 | 95.80 |
| sinpos+ft | | 42.80 | 93.57 | 96.08 | 95.19 | 38.91 | 91.85 | 95.08 | 93.80 | 33.17 | 85.54 | 90.08 | 88.28 | 44.75 | 94.76 | 96.98 | 95.68 |
| CAPE+ft, $\lambda = 1$ | | 79.26 | 94.15 | 96.14 | 95.47 | 75.63 | 92.86 | 95.04 | 94.02 | 66.43 | 86.84 | 90.62 | 89.13 | 81.36 | 95.39 | 96.94 | 95.96 |
| CAPE+ft | | 79.99 | 93.82 | 96.06 | 95.32 | 76.09 | 92.31 | 95.24 | 94.16 | 67.00 | 85.91 | 90.04 | 88.75 | 81.70 | 95.02 | 96.97 | 96.09 |
| UniViT, sinpos | mix | 94.22 | 95.40 | 96.04 | 95.96 | 92.93 | 94.43 | 94.93 | 94.76 | 86.55 | 88.80 | 89.98 | 89.93 | 95.28 | 96.31 | 96.73 | 96.65 |
| UniVit, CAPE $\lambda = 1$ | | 94.39 | 95.56 | 96.18 | 96.02 | 93.02 | 94.48 | 95.23 | 95.19 | 86.72 | 88.91 | 90.30 | 90.33 | 95.29 | 96.40 | 97.00 | 96.84 |
| UniViT, CAPE | | 94.35 | 95.44 | 96.04 | 95.72 | 92.92 | 94.36 | 95.11 | 94.69 | 86.76 | 89.18 | 90.45 | 89.68 | 95.50 | 96.42 | 96.87 | 96.37 |
| abspos-S | $224^2$ | 92.13 | 94.69 | 94.09 | 85.75 | 90.78 | 93.74 | 92.97 | 81.28 | 83.44 | 87.62 | 86.72 | 73.72 | 93.82 | 95.95 | 95.30 | 85.73 |
| UniViT-S, CAPE, $\lambda = 1$ | mix | 92.98 | 94.64 | 95.53 | 95.35 | 92.08 | 93.98 | 94.65 | | 85.15 | 87.92 | 89.64 | 89.02 | 94.88 | 96.00 | 96.72 | 96.37 |
| abspos-Ti | $224^2$ | 86.54 | 90.93 | 90.16 | 80.80 | 85.59 | 90.13 | 88.48 | 76.10 | 76.50 | 82.39 | 81.41 | 68.03 | 89.74 | 93.25 | 91.81 | 81.00 |
| UniViT-Ti, CAPE, $\lambda = 1$ | mix | 86.94 | 89.74 | 91.33 | 90.80 | 86.03 | 88.79 | 90.35 | 89.25 | 77.33 | 80.97 | 83.42 | 82.41 | 90.17 | 92.31 | 93.44 | 92.37 |

For all ViT/UniViT models presented in Figures 2 and 3, and in the ablation study below, we report their top-1 and top-5 accuracies in Tables 3 and 4, respectively, evaluated on the ImageNet validation and ImageNet-v2 test sets and on images with different resolutions.

We train models in Flashlight framework[6] where ViT/DeiT [47] training is reproduced following an original implementation;[7] initialization is set to the truncated normal distribution, Rand-Augment [10], Mixup [56] and Cutmix [54], random erasing [58] and repeated augmentation [25, 4] are used as data augmentations; training is done with AdamW optimizer for 300 epochs. All models use learnable absolute positional embedding for the class token. We train ViT (UniViT) models on 16 GPUs, V100 32GB, with mixed-precision and batch size 64 per GPU for 19-56h (37h) depending on the input resolution ($384^2$ resolution is trained on 32 GPUs with batch size 32 per GPU).

In Figure 4 the throughput is measured as the number of images that can be processed per second on single 16GB V100 GPU following the benchmarking method from [47]: for each image resolution we pass the largest possible batch size and calculate the average time over 30 runs to process that batch.

## B.2 Finding the Best Resolution for UniViT Evaluation

In this section we describe an evaluation procedure that improves UniViT with CAPE performance by resizing input images to an optimal resolution. We split ImageNet validation images into 8 bins, according to their size $s = \min(h, w)$, where $h$ and $w$ are image height and width, respectively: $s \in [54, 100]$, $s \in [101, 150]$, $s \in [151, 200]$, $s \in [201, 250]$, $s \in [251, 300]$, $s \in [301, 350]$, $s \in [351, 384]$, $s \in [385, \inf]$. For each bin we consider several resizing strategies: i) resize all images in a bin either to $160^2$, or $224^2$, or $384^2$; ii) resize all images to the minimum $s$ value in a bin, *Min*; iii) resize all images to the maximum $s$ value in a bin, *Max*; iv) use image's original size but still perform a central *rectangular* crop in a similar manner as standard evaluation is done for ImageNet, *Original*; v) use image's original size, *Original (no crop)*. For the bin with high resolution images $[385, \inf]$ we use $500^2$ as the maximum resize value and apply neither iv) nor v) strategies as there are images with $s > 5000$ px. We report top-1 accuracy for this evaluation procedure with different strategies per each bin in Table 5: for the best strategy in each bin (table row) we report accuracy (%) while for other strategies in the same bin we report absolute drop in accuracy compared to the best value. Best values are additionally marked in bold.

Table 5: UniViT with CAPE evaluation on ImageNet validation set with different strategies on input resizing. Images are split into 8 bins by minimum spatial size. We report best top-1 accuracy (%) in each row (bold) and the drop in accuracy compared to this best accuracy for other columns.

| # Images | Min Res. | Max Res. | $160^2$ | $224^2$ | $384^2$ | Min | Max | Original | Original (no crop) |
|---|---|---|---|---|---|---|---|---|---|
| 146 | 54 | 100 | **84.25** | -0.69 | -0.69 | -19.86 | -1.37 | -6.16 | -8.22 |
| 221 | 101 | 150 | -0.81 | -1.36 | -0.45 | -5.43 | -2.71 | -5.43 | **74.66** |
| 372 | 151 | 200 | -1.61 | -1.08 | -2.42 | -3.23 | -2.42 | **78.49** | -1.61 |
| 538 | 201 | 250 | -3.16 | -1.30 | -1.49 | -2.60 | -1.86 | **81.97** | -2.42 |
| 1090 | 251 | 300 | -4.40 | -1.47 | -0.28 | -1.56 | -1.01 | **79.72** | -0.37 |
| 8496 | 301 | 350 | -3.83 | -1.88 | -0.48 | -1.39 | -1.08 | **83.57** | -0.48 |
| 24538 | 351 | 384 | -4.14 | -1.88 | -0.43 | -0.79 | -0.43 | **82.10** | -0.13 |
| 14599 | 385 | - | -3.30 | -1.25 | -0.14 | -0.13 | **84.46** | - | - |

## B.3 Visualization of Positional Embeddings

We visualize positional embeddings (excluding class token) for ViT models trained on $224^2$ input resolution in Figure 9; for each positional embedding we plot every 20th among 768 components (each row corresponds to a particular component) and visualize its values with the image of shape $(r/16)^2$ where $r^2$ is an input image resolution. We consider input resolutions $160^2$, $224^2$ and $384^2$ shown as left, middle and right sub-columns for each positional embedding in Figure 9.

---

[6] https://github.com/flashlight/flashlight

[7] https://github.com/facebookresearch/deit

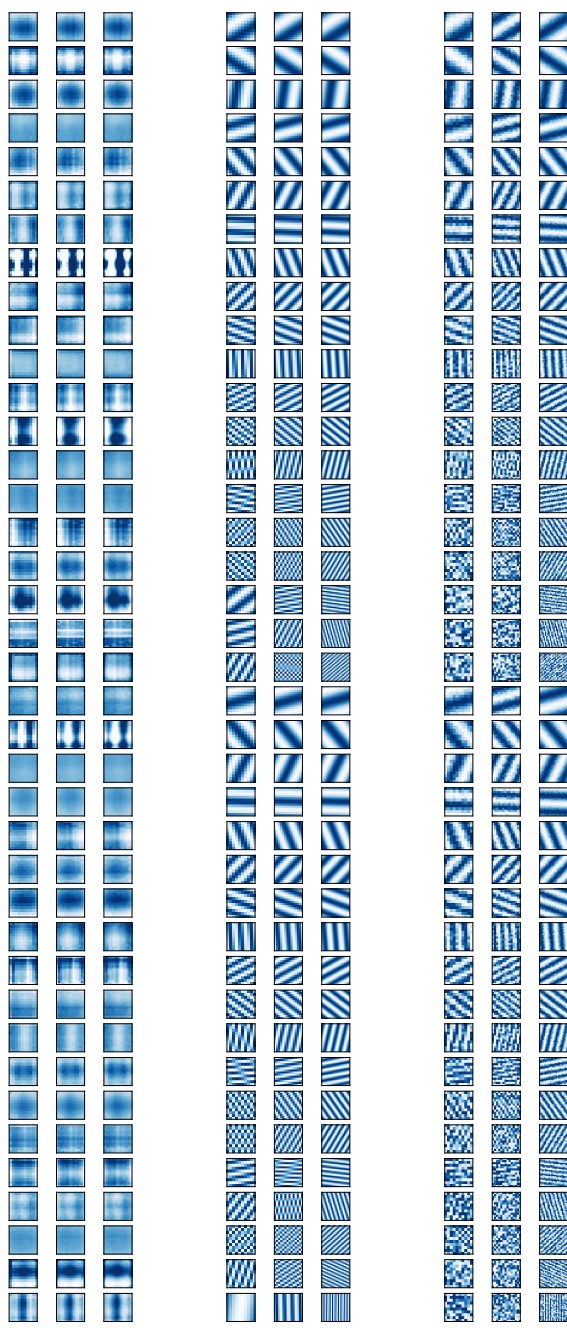

Figure 9: Visualization of positional embeddings for ViT models trained on $224^2$ resolution: *abspos* (left), *sinpos* (middle) and CAPE (right). Each column consists of 3 sub-columns corresponding to input resolutions $160^2$, $224^2$ and $384^2$. Only some components (each 20th out of 768) are shown. When *sinpos* applied to low-resolution images, spacial aliasing is visible in latest components of embeddings. CAPE's augmentations destruct this patterns and prevent model from over-fitting.

## B.4 Ablations

Both *sinpos* and CAPE first re-scale patch positions $(x, y)$ to the $[-1, 1]$ interval independently from the image resolution. We study if an alternative re-scaling of patch positions during inference improves performance on resolutions other than training ones. For ViT models trained with either *sinpos* or CAPE on $224^2$ resolution we perform evaluation on $r^2 = 160^2$ and $r^2 = 384^2$ resolutions

by re-scaling $(x, y)$ to $[-\gamma, \gamma]$: $\gamma$ is set to either 1 (baseline strategy), $r/224$, or $\sqrt{(r/224)}$. Results of this comparison, Table 6, are consistent across models and suggest that for applying to smaller resolution ($160^2$) decreasing scale $\gamma$ to match density of patches on a plane to train-time is beneficial; however, opposite effect is observed when model is applied to higher resolution ($384^2$) inputs, potentially because distances between patch positions in this case were not observed at training time. For simplicity we use re-scaling to $[-1, 1]$ in the rest of our experiments.

Table 6: Ablation study on re-scaling positions to the range $[-\gamma, \gamma]$ for ViT models trained on $224^2$ images with *sinpos* or CAPE. We report top-1 accuracy (%) on ImageNet validation set evaluated on images scaled to $160^2$ and $384^2$ resolutions.

| Model | $\gamma$ | Top-1, $r = 160$ | Top-1, $r = 384$ |
|---|---|---|---|
| | $r/224$ | 77.89 | 72.01 |
| sinpos | $\sqrt{r/224}$ | 77.11 | 76.94 |
| | 1 | 77.15 | 79.72 |
| | $r/224$ | 77.96 | 80.18 |
| CAPE, $\lambda = 1$ | $\sqrt{r/224}$ | 77.80 | 80.51 |
| | 1 | 77.70 | 80.38 |
| | $r/224$ | 77.28 | 80.20 |
| CAPE | $\sqrt{r/224}$ | 77.18 | 80.28 |
| | 1 | 77.14 | 80.33 |

In Figure 10, we compare *sinpos* and CAPE for ViT models in more detail. Overall, CAPE performs better or similar to *sinpos* on the training resolution while it significantly outperforms *sinpos* on other resolutions. In Figure 11 we study the importance of scale $\lambda$ for CAPE in ViT models. Scale $\lambda_{max} > 1$ slightly improves generalization for higher and lower resolutions.

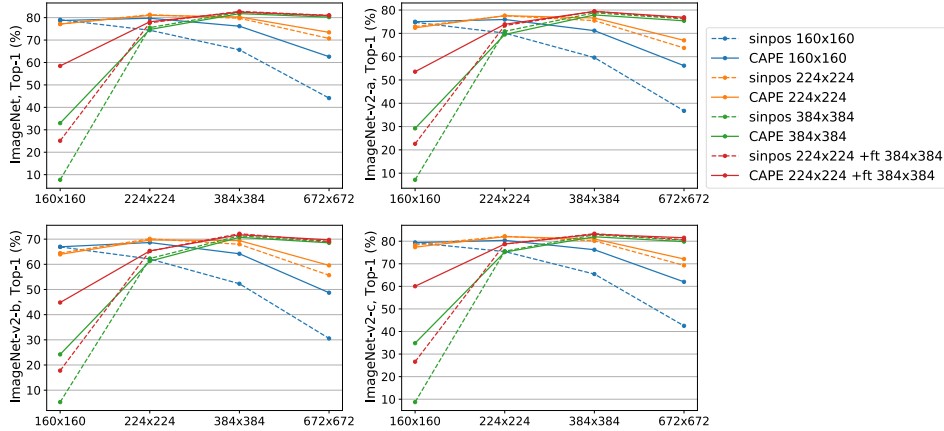

Figure 10: Comparison of top-1 accuracy between *sinpos* and CAPE trained on either $160^2$, or $224^2$, or $384^2$ resolutions and evaluated across the board. Models trained on $224^2$ resolution and further fine-tuned on $384^2$ resolution are marked with "+ft".

In Figure 12 we study the importance of global $\Delta$ and local $\epsilon$ shifts for CAPE in ViT models trained on $224^2$ resolution. On higher resolutions, $384^2$ and $672^2$, models with both shifts (solid) perform similar or better than models trained with either local (dotted-dashed) or global (dashed) shifts. Overall, only one of the shifts, global or local, can be used while the most important CAPE's parameter is the global scale. On the other hand, any combination of augmentations in CAPE clearly outperforms *sinpos* on resolutions different from training one.

In Figure 13 we study if CAPE's augmentations are beneficial for UniViT model, compared to using UniViT with *sinpos*. Overall *sinpos* performs worst among UniViT models, while outperforming UniViT with CAPE and global scaling on $672^2$ resolution. UniViT with CAPE and no global scaling ($\lambda_{max} = 1$) performs the best on all resolutions, suggesting that variability in training resolutions provides a sufficient base for generalization to higher resolutions.

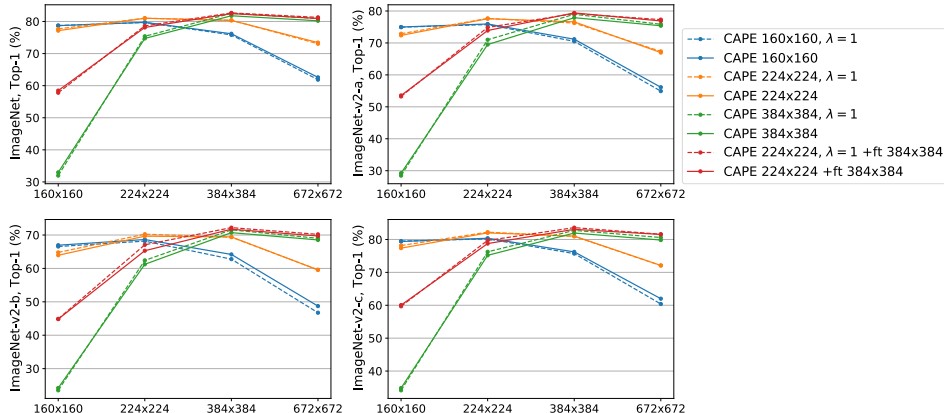

Figure 11: Comparison of top-1 accuracy for CAPE with $\lambda_{max} = 1$ (dashed) and $\lambda_{max} = 1.4$ (solid) trained on either $160^2$, or $224^2$, or $384^2$ resolutions and evaluated across the board. Models trained on $224^2$ resolution and further fine-tuned on $384^2$ resolution are marked with "+ft".

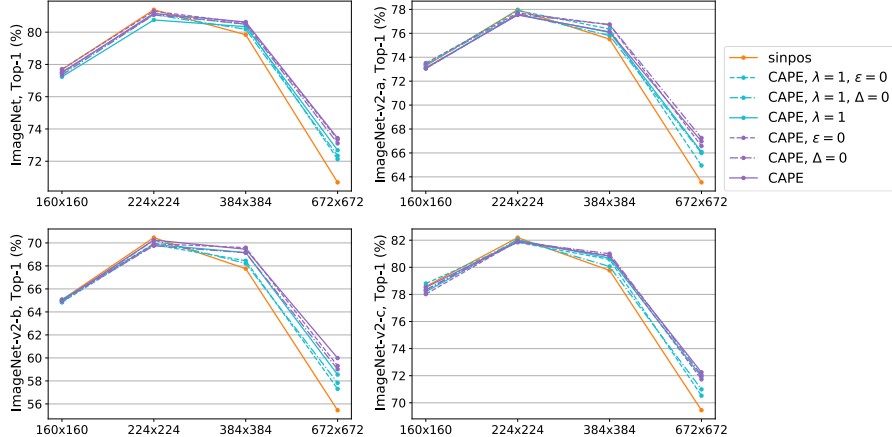

Figure 12: Comparison of top-1 accuracy between *sinpos* and CAPE with different configurations on global, local shifts and global scaling trained on $224^2$ resolution.

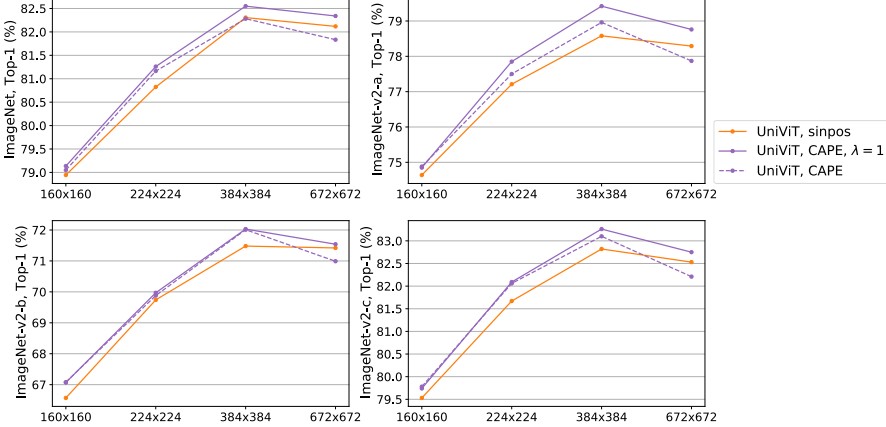

Figure 13: Comparison of top-1 accuracy between *sinpos* and CAPE (with and without global scaling) trained on the mixture of resolutions $\{128^2, 160^2, 192^2, 224^2, 256^2, 288^2, 320^2\}$.

## C Automatic Speech Recognition Experiments

### C.1 Data

For WSJ data we consider the standard subsets *si284*, *nov93dev* and *nov92* for training, validation and test, respectively. We remove any punctuation tokens from *si284* transcriptions before training. TED-LIUM v3 dataset is based on TED conference videos. We use the last edition of the training set (v3); validation and test sets are kept consistent (and thus numbers are comparable) with the earlier releases. We follow the Kaldi recipe [39] for data preparation. In Tables 7 and 8 we present statistics of the datasets used in Section 5.2. One could notice that TED-LIUM v3 validation and test sets have samples with significantly longer duration and larger number of words in their transcriptions, which makes these sets the most challenging among other public data.

Table 7: Statistics on datasets: sampling frequency, duration (in hours), and speech type.

| Data | kHz | Train (h) | Valid (h) | Test (h) | Speech |
|------|-----|-----------|-----------|----------|--------|
| WSJ  | 16  | 81.5      | 1.1       | 0.7      | read    |
| TL   | 16  | 452       | 1.6       | 2.6      | oratory |
| LS   | 16  | -         | -         | 5.4+5.4  | read    |
| RV   | 16  | -         | -         | 18.8+19.5 | diverse |

Table 8: Statistics on datasets: mean sample duration (in seconds) and mean sample transcription length (in words).

| Data | Train $\mu \pm \sigma$ (s) | Valid $\mu \pm \sigma$ (s) | Test $\mu \pm \sigma$ (s) | Train $\mu \pm \sigma$ (wrd) | Valid $\mu \pm \sigma$ (wrd) | Test $\mu \pm \sigma$ (wrd) |
|------|------|------|------|------|------|------|
| WSJ | $7.8 \pm 2.9$ | $7.8 \pm 2.9$ | $7.6 \pm 2.5$ | $17 \pm 7$ | $16 \pm 7$ | $17 \pm 6$ |
| TL | $6 \pm 3$ | $11.3 \pm 5.7$ | $8.1 \pm 4.3$ | $17 \pm 10$ | $35 \pm 20$ | $24 \pm 15$ |
| LS | - | - | $7 \pm 4.8$ | - | - | $19 \pm 13$ |

### C.2 Acoustic Model Training

For all experiments we compute 80 log-mel spectrogram features for a 25ms sliding window, strided by 10ms (unless we explicitly vary STFT hop distance). All features are normalized to have zero mean and unit variance per input sequence before feeding into the neural network.

The self-attention dimension is 768 and the feed-forward network (FFN) dimension is 3072 in each Transformer layer. We use dropout $0.3$ after the convolution layer; for all Transformer layers, we use dropout on the self-attention and on the FFN, and layer drop [19], dropping entire layers at the FFN level. Transformer dropout and layer drop values are set to be $0.4$ for WSJ and $0.1$ for TED-LIUM v3 training.

SpecAugment [37] is used for data augmentation during training: there are two frequency masks, and ten time masks with maximum time mask ratio of $p = 0.1$, the maximum frequency bands masked by one frequency mask is 30, and the maximum frames masked by the time mask is 50; time warping is not used. We use the Adagrad optimizer [17]. All models are trained with dynamic batching (average batch size is 240s/GPU) and mixed-precision computations on 16 GPUs (Volta 32GB) for 1 day on WSJ and 3-4 days on TED-LIUM v3. All ASR experiments are done within Flashlight framework on top of the publicly available training configurations[8] for baselines with *relpos* from [30].

### C.3 Padding-free ASR with CAPE and Variable STFT Hop Distance

We have implemented pipeline where padding is no longer used to form a batch from samples with different input duration. For each audio in the batch short-time Fourier Transform (STFT) hop distance $H$ is set in a way that output number of frames is the same (except rounding) as for hypothetical audio which has duration equal to the mean over batch and is processed with $H = 10$ms. Because

---

[8] https://github.com/flashlight/wav2letter/tree/master/recipes/rasr

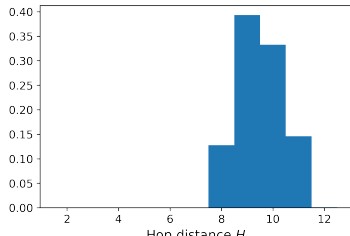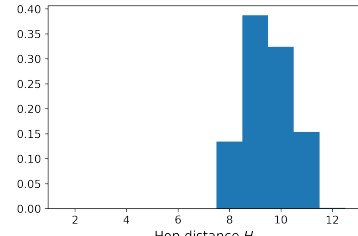

Figure 14: Hop distance distribution for WSJ (left) and TED-LIUM v3 (right) data.

the hop distance is an integer number, number of frames after STFT is matched only approximately within a batch, so we reduce the number of frames in each sample to match the shortest sample in the batch by randomly and uniformly skipping frames. To have low variation of samples duration in a batch (which implies limited vatiation in $H$) the following shuffling strategy is performed for every epoch: i) compute perturbed sample duration by multiplying original sample duration by a random number from $\mathcal{U}(0.85, 1.15)$; ii) sort samples by their perturbed duration; iii) batches are formed by grouping sequential samples. Example of hop distance distribution after proposed shuffling strategy for WSJ and TL data is shown in Figure 14. For both *sinpos* and CAPE embeddings we train models with this new pipeline and observe mostly lower WER and improved generalization, especially on TL test and RV data which are the most challenging among evaluation sets, Figures 15 and 16.[9]

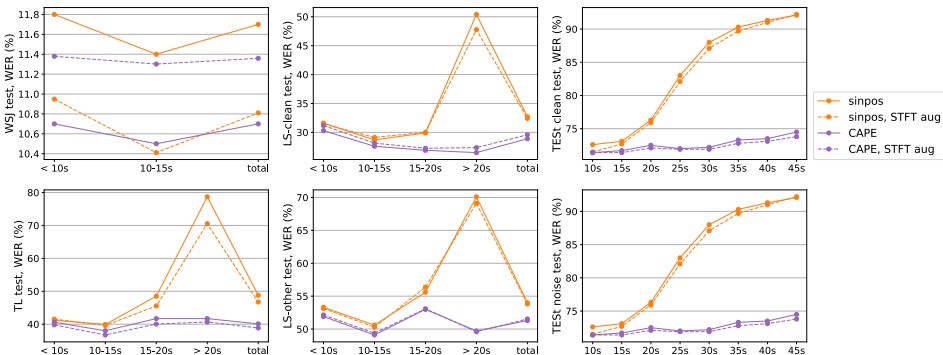

Figure 15: Word error rate comparison for models trained on WSJ data with *sinpos* or CAPE ($\lambda = 1$) with classical pipeline (solid) or with variable STFT hop distance (dashed).

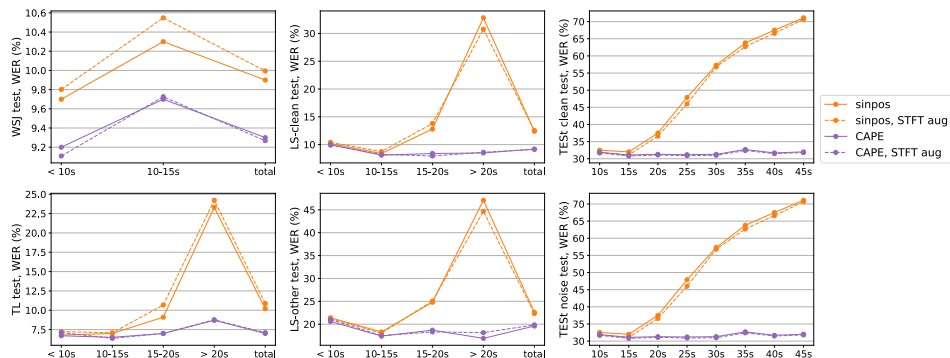

Figure 16: Word error rate comparison for models trained on TED-LIUM v3 data with *sinpos* or CAPE ($\lambda = 1$) with classical pipeline (solid) or with variable STFT hop distance (dashed).

---

[9]For all models evaluation the batch size is set to 1 and $H = 10$ms, thus padding never affects the performance on validation and test sets.

As discussed in Section 5.2, STFT hop distance can be viewed as image resolution in vision and padding-free ASR – as UniViT. By adjusting hop distance during inference for padding-free ASR we can achieve higher throughput with similar recognition quality, see Figure 17. Moreover, it is robust to STFT hop distance variation having almost the same WER for $H \in [5, 12]$ms and less performance degradation for $H > 12$ms.

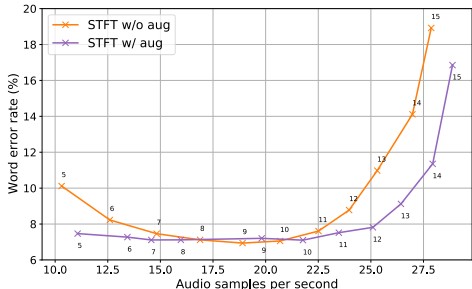

Figure 17: Dependence between throughput and word error rate on TED-LIUM v3 test set for models trained with CAPE ($\lambda = 1$) on TED-LIUM v3 data: with classical pipeline (orange) and with variable STFT hop distance (purple). Different throughput values correspond to different STFT hop distances in ms (crosses) used during evaluation.

## C.4  Ablations

First, we study dependence between the global shift value $\Delta_{max}$ and model's performance and generalization abilities to the long duration. Varying the global shift we observe in Figure 18 that larger global shift leads to a better generalization on longer audios, so that CAPE with 30-60s global shifts is able to process 45s audio with the same performance as 10s on RV data.

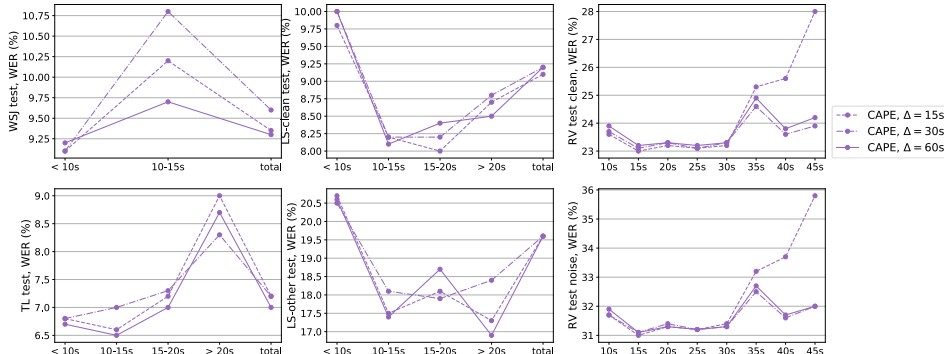

Figure 18: Word error rate comparison for models trained on TED-LIUM v3 data with CAPE and different global shifts which cover 15, 30 or 60s. The global scale is set to $\lambda_{max} = 1$.

Secondly, we study the necessity of the local shift in CAPE. In Figure 19 we observe that local shift absence hurts the performance and generalization across the board.

Thirdly, we study the necessity of the global scaling in CAPE. In Figure 20 we observe that for WSJ models global scaling hurts a bit performance on public data while performs and generalizes better for RV data. In contrast, for TL models we observe in Figure 21 that overall the global scaling improves performance on public data while hurts performance on RV data. Thus, the global scaling should be tuned separately depending on the data type.

As an ablation study we perform additional experiments with *relpos*. First, we restrict *relpos* context to small duration, 6s, to prevent over-fitting to relative positions: *relpos 6s* outperforms *relpos* on both public and RV data for both models trained on WSJ and TL having significantly better generalization to long audio durations. *Relpos 6s* is performing similar to *abspos* for duration $> 20$s while CAPE still

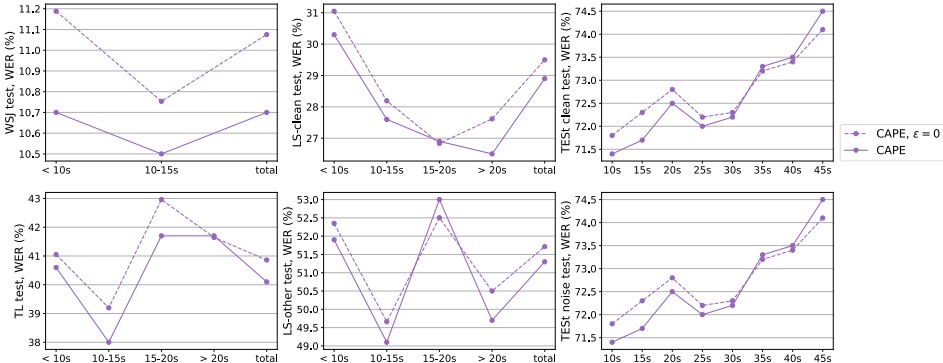

Figure 19: Word error rate comparison for CAPE models trained on WSJ data with global shift only (solid) or with global and local shifts together (dashed). The global scale is set to be $\lambda_{max} = 1$.

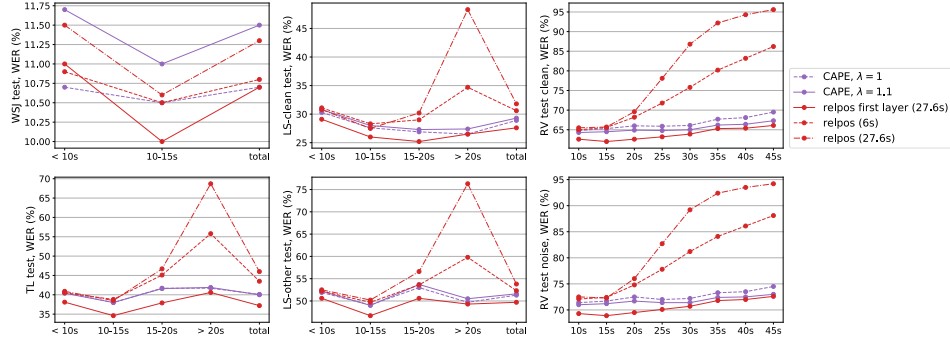

Figure 20: Word error rate comparison for models trained on WSJ data with different positional embeddings. *Relpos first layer* refers to a model where *relpos* is used only in the first Transformer layer with 27.6s context to the left/right and no other positional embeddings are used.

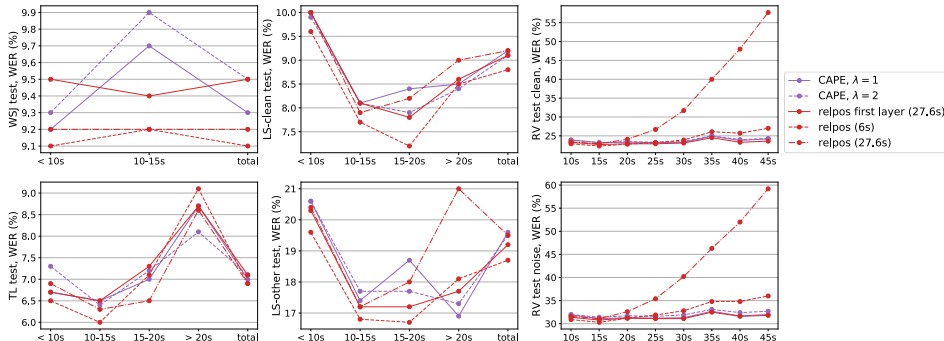

Figure 21: Word error rate comparison for models trained on TED-LIUM v3 data with different positional embeddings. *Relpos first layer* refers to a model where *relpos* is used only in the first Transformer layer with 27.6s context to the left/right and no other positional embeddings are used.

outperforms *relpos 6s* on $> 20$s, Figures 20 and 21. Independently, we also tried sinusoidal relative positional embedding [11] but it performed worse than learnable relative positional embedding.

Second, having in mind that *nopos* performs well for a model trained on TL and CAPE's ability to learn spatial relations using a single layer, we wonder if *relpos* should be used only in the first Transformer layer (in literature *relpos*, when used, is applied in every attention layer). We modify *nopos* model by injecting *relpos* embedding (27.6s context to the right/left) only in the first Transformer layer: no any other Transformer layers use any positional embeddings, Figures 20 and 21.

This *relpos first layer* model behaves surprisingly well: for a WSJ model it outperforms CAPE on both public and RV data; for a TL model it behaves similar to CAPE on public data and a bit better on RV data. Both CAPE and *relpos first layer* have similar mostly uniform performance profiles across different audio durations on RV data. This observation asks for reconsidering the standard usage of positional embedding for CTC-based models in speech recognition.

## C.5 No Positional Embedding Discussion

As demonstrated above in Figure 6, *nopos* performs similar to different positional embeddings on both public and RV data while having reliable generalization to the long audio fragments. We figured out that the key components of this phenomenon and *nopos* success are i) enough training data; ii) sufficient model capacity and iii) CTC loss.

For the first point we saw that *nopos* model trained on WSJ, a 5x smaller dataset than TL, performs poorly having 45-50% WER even on in-domain data. For the second point we perform an additional ablation on WSJ data by decreasing dropout and layer drop in each Transformer layer from 0.4 to 0.1: with increased model capacity *nopos* reduces the WER by 30% and gets closer to other positional embeddings, Figure 22. For the third point we perform another ablation by comparing with sequence-to-sequence (seq2seq) training: we use exactly the same encoder $\mathbf{H}^{L_e}$ (with various positional embeddings) but replace last linear layer and CTC loss with the decoder, encoder-decoder attention, and cross-entropy loss where the probability distribution of the transcription is factorized as

$$p(y_1, ..., y_n) = \prod_{i=1}^{n} p(y_i \mid y_0, ..., y_{i-1}, \mathbf{H}^{L_e})$$

where $y_0$ is a special symbol indicating the beginning of the transcription. The decoder is a stack of 6 Transformer layers with encoding dimension 256, learnable relative positional embedding with 9.6s left-only context and 4 attention heads. Dropout and layer drop in the decoder layers are set to 0.2.

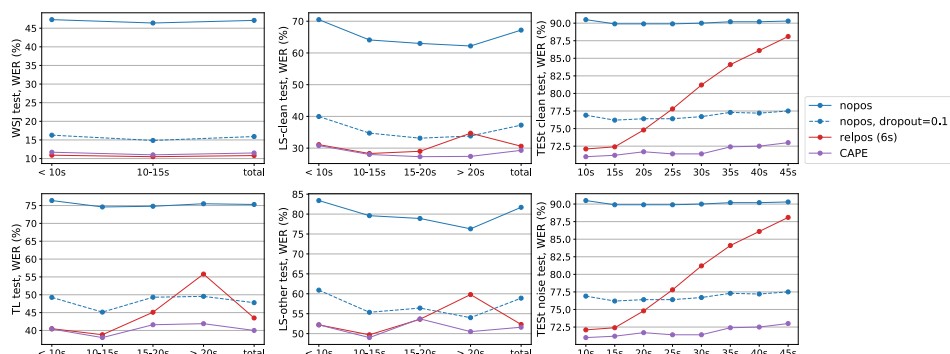

Figure 22: Word error rate comparison for models trained on WSJ data with different positional embeddings. Baseline models, *nopos* and CAPE, use 0.4 dropout and 0.4 layer drop in every Transformer layer, while *nopos, dropout=0.1* uses 0.1 for both values.

In Figure 23 we show comparison between CTC and seq2seq models trained on TL with either *nopos* or *relpos* in the encoder.[10] Seq2seq *nopos* performs significantly worse than seq2seq *relpos* and moreover has higher WER variation on validation data. This result is opposite to the *nopos* CTC-based training, suggesting that CTC loss is able to train with enough data or model capacity and no positions provided.

These observations only partially overlap with known results: for seq2seq training it was shown recently that relative positions can be modeled via a deep stack of convolutional layers in the encoder [33] or via convolutions inserted directly in each encoder's Transformer layer [57]. In contrast to the listed works, our encoder has vanilla Transformer layers and only one convolutional layer at the beginning. Thus, *nopos* model has very limited context to model relative positions, which affects seq2seq training (it has to "locate" corresponding timepoint in audio with attention) more than

---

[10]Encoder remains the same for both CTC and seq2seq models; for seq2seq models decoder is also identical.

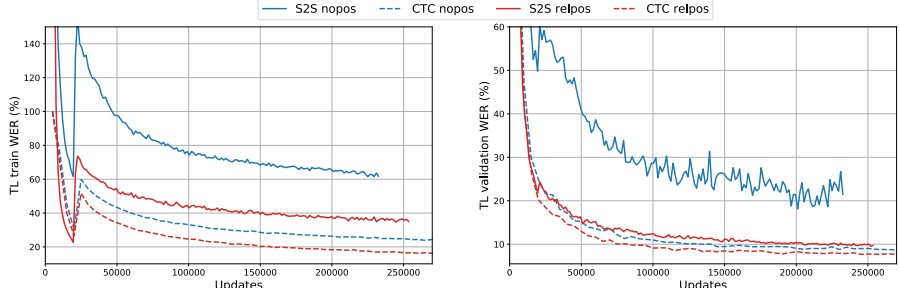

Figure 23: Word error comparison for CTC and seq2seq models trained on TED-LIUM v3 data without any positional embedding in the encoder (*nopos*) or with learnable relative positional embedding in every encoder-Transformer layer (*relpos*).

CTC-based, which uses explicit time ordering. In line with this interpretation, for the hybrid ASR systems dropping positional information does not drive to significant deterioration of quality [51].

# D   Machine Translation

## D.1   Technical Details

For machine translation experiments we have implemented CAPE within ADMIN's [31, 32] open-sourced code[11] which is based on Fairseq[12] toolkit [35]. We precisely follow open-sourced recipes for ADMIN with *sinpos*[13] including data preparation step; the only change we introduce is usage of different positional embeddings. Besides, we do not perform model averaging as was done in [31, 32].

All English-German (DE) models are trained for 150 epochs on 4 GPUs (Volta V100 16GB) for 30h (6l-6L) or 70h (18L-18L). English-French (FR) 6L-6L models are trained for 75 epochs on 8 GPUs (Volta V100 16GB) for 60h. Each configuration is trained starting from 3 different random seeds.

As mentioned in Section 5.3 we scale positions of source language by a factor $\alpha$ computed based on train data statistics only as $\alpha = \frac{\text{\# tokens in target corpus}}{\text{\# tokens in source corpus}} \in \mathbb{R}$: it is set to $\alpha = 1.0337$ for DE and $\alpha = 1.1632$ for FR. For all machine translation experiments with CAPE we skip the mean-normalization step to have source and target sentences aligned at the first position and prepend "begin of sentence" in the source sentences to give a hint of first position for the model's decoder (as after global shift there is no way to determine the first position from its positional embedding anymore). Additionally, we do not apply any global scaling. For the global shift we sweep values 5, 10 and 50 while the local shift is set to maximum to preserve positions order, $\epsilon_{max} = 0.5$.

Table 9: Slowdown for relpos / no relpos. (Top) 1000 context left/right, (Bottom) 100 context left/right. Run on V-100 GB32, 100 runs, 10 runs warmup, batch 50, emb 768, head 8.

| Model | FP16 Len-10 | FP32 Len-10 | FP16 Len-100 | FP32 Len-100 | FP16 Len-1000 | FP32 Len-1000 |
|---|---|---|---|---|---|---|
| Transformer layer | 2.1 | 2.3 | 3.3 | 2.3 | 2.2 | 1.7 |
| Encoder | 2.2 | 2.2 | 2.6 | 1.7 | 2.0 | 1.6 |
| Decoder | 1.8 | 1.7 | 1.9 | 1.4 | 1.6 | 1.4 |
| Transformer layer | 2.1 | 2.1 | 1.9 | 1.2 | 1.4 | 1.2 |
| Encoder | 2.3 | 2.4 | 1.5 | 1.1 | 1.4 | 1.2 |
| Decoder | 1.7 | 1.7 | 1.2 | 1.1 | 1.2 | 1.1 |

---

[11] `https://github.com/LiyuanLucasLiu/Transformer-Clinic`

[12] `https://github.com/pytorch/fairseq`

[13] `https://github.com/LiyuanLucasLiu/Transformer-Clinic/blob/master/nmt-experiments:` `wmt14_en-de.md` and `wmt14_en-fr.md`

## D.2 Computational Cost

For machine translation experiments we estimate the forward and backward time slowdown when learnable relative positional embedding is used in Transformer layer compared to vanilla Transformer layer. Besides we evaluate slowdown for entire encoder and decoder. Forward and backward time benchmarking is done in Fairseq (PyTorch) for different input sequences length (10, 100, 1000), for different context size of relative positional embedding (100 and 1000 tokens to left/right) in full-precision (fp32) and half-precision (fp16) floating-point computations. We use Volta V-100 32GB GPU and report slowdown for average time of forward and backward passes measured across 100 updates with 10 updates of warmup, see Table 9. The batch size is set to 50, attention heads are set to 8 and embedding dimension is 768 (typical use case), there are 6 Transformer layers in encoder and decoder. Results in Table 9 demonstrate that relative positional embedding indeed has additional computational cost which is even more pronounced with mixed-precision computations.