# OpenReview forum: "CAPE: Encoding Relative Positions with Continuous Augmented Positional Embeddings"
_NeurIPS.cc/2021/Conference — NeurIPS 2021 Poster_

### Official Review · Reviewer_b4uj · 2021-07-11

**Rating:** 4
**Confidence:** 4

**Summary:**

This paper claimed that the existing positional encoding mechanism (i.g., absolute or relative positional encodings) for the Transformer model has a generalization issue, that is, these position encoding methods have their own advantages and disadvantages, which may prevent the Transformer model from simulating positional information. Through formal analysis of the existing position encoding mechanisms, the authors proposed an augmentation-based approach to integrating the advantages of the existing absolute position encoding and relative position encoding. It is expected that the positional information can be efficiently and simply modeled for the Transformer model. Finally, the proposed method was evaluated on several tasks which adapted to the Transformer model.

**Ethics Review Area:**

["I don’t know"]

**Main Review:**

1. There is a lack of sufficient argument support and analysis about whether this claimed generalization issue really exists.

2. The Transformer model was originally proposed for machine translation tasks. Positional encoding plays an important role. However, the improvements over the baseline on machine translation tasks are marginal over the baseline abspos in Table 2.

3. There is a lack of necessary comparisons with relative positional encoding. I am afraid that CAPE may be inferior to the existing relative positional encoding.

4. There is a lack of necessary ablation experiments to verify that this marginal improvement indeed comes from the alleviation of the claimed generalization issue.

5. If all model parameters in the trained baseline Transformer model are frozen except the positional embeddings, the trained baseline Transformer model is further fine-tuned by one or several Epochs. Or, in the beginning, these positional embeddings are frozen from the beginning and are applied to different position encoding methods to verify their effectiveness.

Overall, I think the motivation of this article is unclear, its novel is limited, its improvement is marginal on the machine translation tasks.

Update:

The author's response addressed part of my main concerns, and I will increase my score from 3 to 4.

**Time Spent Reviewing:**

6 hours

---

> ### Author Response · Authors · 2021-08-10
> **First comments/explanations**
>
> We thank a lot reviewer b4uj for time spent reviewing our paper and helpful comments/questions. Please find our detailed comments and explanations below.
>
> > 1. There is a lack of sufficient argument support and analysis about whether this claimed generalization issue really exists.
>
> For the **vision domain** we analyse the recent state-of-the-art model ViT and with figures 2 and 3 we demonstrate the generalization issue for resolutions different from training one, especially for low (160x160) or high resolutions (672x672). Poor transfer of model to different resolution is discussed in DeiT paper https://arxiv.org/abs/2012.12877. With figures 2 and 3 we have demonstrated that CAPE is able to deal with the issue and improve generalization performance.
>
> For the **speech domain** we analyse state-of-the-art models trained with CTC loss for two datasets with different scales of available training data (80h and 450h). We evaluate each of these two models on 6 different datasets for different audio durations. With Figures 5 and 6 we demonstrate that if models are trained on short durations < 20s they have a generalization issue on durations >20s for abspos, sinpos and even relpos while CAPE is able to deal with the issue and improve generalization performance.
>
> Generalization to longer sequences is a widely known issue. Related works are mentioned in the introduction (rows 20-32), for example, study on the generalization issue and possible remedy music generation https://openreview.net/pdf?id=rJe4ShAcF7, speech recognition https://arxiv.org/pdf/1911.00203.pdf, machine translation https://www-i6.informatik.rwth-aachen.de/publications/download/1132/Rosendahl-IWSLT-2019.pdf (this last reference will be added into the final text too).
> Also we refer to the recent overview of positional embedding in transformers where the generalization issues to long sequences are brought up multiple times https://arxiv.org/pdf/2102.11090.pdf.
> Interestingly, the generalization issue is frequently stated as obvious (e.g. https://arxiv.org/pdf/2009.13658.pdf) and not requiring any experiments to confirm. This generalization argument is used to justify introduction of relative positional embeddings. Our work demonstrates that this is not the only possible remedy.
>
> > 2. The Transformer model was originally proposed for machine translation tasks. Positional encoding plays an important role. However, the improvements over the baseline on machine translation tasks are marginal over the baseline abspos in Table 2.
>
> We took the current SOTA model (for example, see https://paperswithcode.com/sota/machine-translation-on-wmt2014-english-french) and didn't change anything except the positional embedding. With only this change we observe improvement out-of-the box with CAPE embedding.
> We have run the same experiments with two other seeds (so that in total 3 runs for each of the models in Table 2). Results are following:
>
> | Model | Lang | valid BLUE| test BLUE |
> | - | - | - | - |
> sinpos, 6L-6L | DE | 26.88 +/- 0.05 | 27.66 +/- 0.10
> abspos, 6L-6L | DE | 26.68 +/- 0.05 | 27.36 +/- 0.06
> CAPE, 6L-6L | DE | 26.86 +/- 0.13 | 27.89 +/- 0.07
> sinpos, 18L-18L | DE | 27.09 +/- 0.06 | 28.28 +/- 0.28
> abspos, 18L-18L | DE | 27.23 +/- 0.02 | 28.26 +/- 0.22
> CAPE, 18L-18L | DE | 27.17 +/- 0.10 | 28.44 +/- 0.06
> sinpos, 6L-6L | FR | 47.27 +/- 0.03 | 41.13 +/- 0.07
> abspos, 6L-6L | FR | 47.22 +/- 0.03 | 41.21 +/- 0.04
> CAPE, 6L-6L | FR | 47.22 +/- 0.03 | 41.59 +/- 0.03
>
> > 3. There is a lack of necessary comparisons with relative positional encoding. I am afraid that CAPE may be inferior to the existing relative positional encoding.
>
> Below we provide full details on each modality:
> - For the **vision domain** we took the recent state-of-the-art transformer model, which was DeiT model https://arxiv.org/abs/2012.12877 (ViT with absolute positional embedding and improved optimization/training scheme). Original ViT paper https://arxiv.org/pdf/2010.11929.pdf performed some ablations with relative positional embedding, however a) no improvement observed and b) no correct setting which encounters 2d nature of images was proposed. The proper version of 2D relative positional embedding is yet to be introduced and verified by the community, and that's not the goal of this work. Contrary to relpos, CAPE readily works with different types of data.
>
> - We perform rather detailed and in-depth analysis of relative positional embedding in **speech**. We start from a strong relpos-based baseline and demonstrate that (unexpectedly) it poorly generalizes to longer audio fragments. We analyzed potential cures: ignoring distant positions and minimized usage of relpos to only one layer (CAPE's high performance hinted to this experiment design). See Figures 19 and 20 in supplementary material. Independently we have also tried sinusoidal relative positional embedding for speech recognition (as Transformer-XL) but it performed worse than learnable relative positional embedding so only the best performing was included in the paper.
>
> - As for **machine translation**, relative positions are commonly considered to perform better but public leaderboards for common MT benchmarks tell the opposite: from En-Fr WMT14 leaderboard (https://paperswithcode.com/sota/machine-translation-on-wmt2014-english-french) we checked top 3 approaches with external data and top 3 without: none of them uses relative positional encoding. So we used the SOTA approach with available implementation and plugged in our embedding for comparison.
>
> > 4. There is a lack of necessary ablation experiments to verify that this marginal improvement indeed comes from the alleviation of the claimed generalization issue.
>
> No such claim was done in the paper regarding machine translation. We did not have this goal and no experiment setup was developed to check this. WMT'14 benchmark has limited length of test sentences. However, deterioration of translation quality for higher input lengths is vastly observed in translation systems. In the paper, we demonstrate CAPE's ability to deal with the generalization issue by experiments in speech and image recognition.
> Main improvement in MT should come from disentangling position and content embeddings, however, we neither claimed this nor planned to make an additional study for that. The main goal of the MT section is demonstration of the applicability of CAPE ideas to NLP problems, particularly in the "cradle" of transformers.
>
> To resolve doubts about generalization issue in MT and CAPE's ability to deal with it we perform the following evaluation. Consider the test sets in WMT'14 and artificially stack sentences: we stack sentences until 300+ length in target language will be in a sentence (~3000 original sentences are transformed into ~270 sentences with length 300+ tokens each); during stacking dots on junctions were replaced with commas  and first letters were lower-cased of the next stacked sentence to make the translation harder for the models. For such prepared test data we now evaluate log loss for each position. In figure https://ibb.co/hDx7S6Q we plot the average log loss across consecutive positions. It can be seen that for positions > 100 for DE and > 200 for FR CAPE has statistically lower -log loss. For En-Fr task there is a significant gap in log loss.
>
> > 5. If all model parameters in the trained baseline Transformer model are frozen except the positional embeddings, the trained baseline Transformer model is further fine-tuned by one or several Epochs. Or, in the beginning, these positional embeddings are frozen from the beginning and are applied to different position encoding methods to verify their effectiveness.
>
> This incorrectly describes our work: a) all models in all modalities are trained from scratch b) all weights in all models are trainable and never frozen c) architectures and hyper-parameters of all models for each particular comparison for every domain are identical except positional embedding.
>
> For example, we first reproduced ViT model training with absolute positional embedding. We then replace the absolute positional embedding in the model architecture with CAPE and train the model from scratch in the same way (number of layers, architecture, optimization, scheduling, augmentations) as the baseline. We thus verify that CAPE works better even with an optimization scheme which was tuned for baseline, not for CAPE.
>
> Similarly for the MT task we take an already tuned recipe (model, initialization, optimizer, lr schedule, tokenization, etc.) for SOTA approach and change only embeddings to demonstrate versatility of our method.

---

> > ### Comment · Reviewer_b4uj · 2021-08-25
> > **The second reviewing**
> >
> > Thanks for your response.
> > Although the authors gave some explanations for my concerns, there are some confused key points:
> >
> > 1. Actually, the generalization issue was claimed in Abstract and Section 3. My understanding is that it is the motivation of this work.
> >
> > 2. Hoping that authors can give the suggested ablation experiments to show that the improvement was indeed from alleviating the claimed generalization issue.
> >
> > 3. The updated experiment confirmed that the improvement was marginal on the machine translation tasks compared to the baseline relative positional encoding.
> >
> > If you can use the corresponding ablation experiments to show the claimed generalization issue and the improvement was indeed from alleviating the claimed generalization issue, I promise will increase my score.

---

### Official Review · Reviewer_MLDf · 2021-07-14

**Rating:** 6
**Confidence:** 3

**Summary:**

This paper introduces a data augmentation method that adds random noise to positions for the purpose of computing position embeddings for Transformers in vision, speech, and machine translation.


**Limitations And Societal Impact:**

I'd like to see (1) better treatment of UniVIT, in the form of (a) more organized motivation and presentation, (b) experiments with UniVIT on all tasks where it is applicable; and (2) significance testing.


**Main Review:**

# Model

The formulation of sinusoidal position encodings as complex numbers is not new; see Wang et al (ICLR 2020), DeBenedetto and Chiang (ICML 2020), Su et al (arXiv:2104.09864).

Can you provide some motivation for your choice of w_{k,x} and w_{k,y}? I’m a little surprised to see a sine/cosine inside another sine/cosine.

Lines 111-112: What is mean-normalization?

# Experiments

The paper presents experiments on three tasks: image recognition, speech recognition, and machine translation.

On image recognition, two different models are tried. Figure 2 shows that CAPE gives slight improvements except when training on 224x224 and, fine-tuning on 384x384 but testing on 160x160, where CAPE outperforms all other methods by a wide margin.

What are the four smaller graphs in Figure 2?

The paper also presents a related method, UniViT, which doesn’t do global scaling; instead, it resizes the images themselves to random sizes. Across all resolutions, this performs consistently well, better than all other methods including CAPE.

On speech recognition (Figure 5-6), CAPE generally does well, especially on their own Robust Video dataset.

On machine translation (WMT14 task), improvements are small (0.27-0.56) and significance testing is not performed.

# Overall

This is a nice, simple data augmentation method that works on several different modalities and doesn't require the computer to be too "creative" (e.g., choosing random words). It seems to help on speech and vision, but not so much on machine translation.

From both a presentational and scientific standpoint, it's confusing that a different method (UniVIT) is introduced that outperforms CAPE. Presentationally, because the paper's title is CAPE and the methods section only describes CAPE, and it's in the experiments section that a new method is unexpectedly introduced. Scientifically, because it's not clear why one does better than the other, and in which scenarios one should prefer one or the other.

**Time Spent Reviewing:**

2

---

> ### Author Response · Authors · 2021-08-10
> **First comments/explanations**
>
> We would like to thank MLDf reviewer for helpful comments/questions, additional references and suggestions on significance test for MT section. Please find our detailed comments and explanations below.
>
> > The formulation of sinusoidal position encodings as complex numbers is not new; see Wang et al (ICLR 2020), DeBenedetto and Chiang (ICML 2020), Su et al (arXiv:2104.09864).
>
> Thanks for pointing out references, we will add them into the text. Just to be clear, we don't state novelty in using complex numbers, rather use them to simplify our analysis.
>
> > Can you provide some motivation for your choice of w_{k,x} and w_{k,y}? I’m a little surprised to see a sine/cosine inside another sine/cosine.
>
> Figure 13 in appendix should be helpful: it visualizes embedding from abspos, sin2d and CAPE. "inner" $\sin k$ and $\cos k$ in the formula are responsible for angle of "hatching" (so all components have different angle, angles uniformly cover possible directions), while "outer" $\sin$ and $\cos$ used in definition of $E_{2k}$ and $E_{2k+1}$ create a wave as in 1d sinusoidal embedding and control hatching density.
>
> This choice was made because of following considerations: a) simple-to-reproduce implementation b) no "selected directions" on the plane, when compared to two 1d embeddings c) embedding is deterministic, no RNG used d) different "hatching densities" allow both precise and approximate positioning.
>
> > Lines 111-112: What is mean-normalization?
>
> For the sequence with length $N$ we compute $\mu_{pos} = \frac{1}{N} \sum_{i=1}^{N} pos_i$ and then we transform every position from $pos_i$ to $pos_i - \mu_{pos}$.
> We will change the parameter name in reference implementation in appendix A from 'normalize' to 'mean_normalize' to make it clear.
>
> > What are the four smaller graphs in Figure 2?
>
> This is a zoomed version of the larger plots of Figure 2, to make a clear (zoomed) view of the region 65-85% accuracy for readability.
>
> > On machine translation (WMT14 task), improvements are small (0.27-0.56) and significance testing is not performed.
>
> We have run the same experiments with two other seeds (so that in total 3 runs for each of the models in Table 2). Results are following:
>
> | Model | Lang | valid BLUE| test BLUE |
> | - | - | - | - |
> sinpos, 6L-6L | DE | 26.88 +/- 0.05 | 27.66 +/- 0.10
> abspos, 6L-6L | DE | 26.68 +/- 0.05 | 27.36 +/- 0.06
> CAPE, 6L-6L | DE | 26.86 +/- 0.13 | 27.89 +/- 0.07
> sinpos, 18L-18L | DE | 27.09 +/- 0.06 | 28.28 +/- 0.28
> abspos, 18L-18L | DE | 27.23 +/- 0.02 | 28.26 +/- 0.22
> CAPE, 18L-18L | DE | 27.17 +/- 0.10 | 28.44 +/- 0.06
> sinpos, 6L-6L | FR | 47.27 +/- 0.03 | 41.13 +/- 0.07
> abspos, 6L-6L | FR | 47.22 +/- 0.03 | 41.21 +/- 0.04
> CAPE, 6L-6L | FR | 47.22 +/- 0.03 | 41.59 +/- 0.03
>
> > From both a presentational and scientific standpoint, it's confusing that a different method (UniVIT) is introduced that outperforms CAPE. Presentationally, because the paper's title is CAPE and the methods section only describes CAPE, and it's in the experiments section that a new method is unexpectedly introduced. Scientifically, because it's not clear why one does better than the other, and in which scenarios one should prefer one or the other.
>
> UniViT is a novel way of training on the mix of resolutions that exploits the benefits of CAPE, but not restricted to CAPE. Because of our formulation of sinusoidal 2D positional embedding instead of absolute positional embedding ViT now can naturally handle multiple resolutions. For UniViT training we test both sinusoidal positional embedding and CAPE, see comparison in the supplementary material in Figure 12. We demonstrate that UniViT + sinpos is worse than UniViT + CAPE. Combination UniViT + CAPE should be preferred due to adjustability and high performance across multiple scales with only 1.1x more training time.
> To remove confusion, labels in the main text will be changed to UniViT + CAPE.
>
> > I'd like to see (1) better treatment of UniVIT, in the form of (a) more organized motivation and presentation (b) experiments with UniVIT on all tasks where it is applicable;
>
> Exploration of broader UniViT applicability to other tasks is a subject of future work.
>
> > and (2) significance testing.
>
> See our comment above with significance testing results.

---

### Official Review · Reviewer_6kLu · 2021-07-16

**Rating:** 6
**Confidence:** 4

**Summary:**

This paper proposes an augmentation-based approach (CAPE) for absolute positional embeddings. The empirical evaluation on state-of-the-art models in machine translation, image and speech recognition demonstrates that CAPE leads to better generalization performance. The paper is cleary-written and the experiments are comprehensive.

**Ethics Review Area:**

["I don’t know"]

**Main Review:**

The paper explains that "Relative positions are more complex to implement and yield inferior model throughput" in the abstract.  It seems that the implementation complexity does not make any sense, since such implementation complexity does not necessarily lead to space or time cost. Such complexity should be clearly stated in space complexity, time complexity, or others that may bring extra cost. It would be better to explain why it affects model throughput. Note that there are some simpler Relative position encoding, see T5 that defined a scalar added to the corresponding logit used for computing the attention weights. This needed furtherly explained, otherwise it may confuse readers.



When discussing complex-valued representation of embeddings in line 81. Please consider the following reference:
Wang et.al. encoding word order in complex embeddings. ICLR 2020.
which also explored the continuous nature of position embeddings.


minor issue:
Figure 1 was not mentioned anywhere in the main text.

**Time Spent Reviewing:**

2 hours

---

> ### Author Response · Authors · 2021-08-10
> **First comments/explanations**
>
> 6kLu, thanks a lot for your time spent reviewing our paper and helpful comments/questions. Detailed comments and explanations are given below.
>
> > The paper explains that "Relative positions are more complex to implement and yield inferior model throughput" in the abstract. It seems that the implementation complexity does not make any sense, since such implementation complexity does not necessarily lead to space or time cost. Such complexity should be clearly stated in space complexity, time complexity, or others that may bring extra cost. It would be better to explain why it affects model throughput. Note that there are some simpler Relative position encoding, see T5 that defined a scalar added to the corresponding logit used for computing the attention weights. This needed furtherly explained, otherwise it may confuse readers.
>
> In the abstract we aimed to point to two things independently: 1) implementation complexity of some relative positional embedding (for example, learnable relative positional embedding and sinusoidal relative positional embedding from Transformer-XL) - this can lead to error-prone code in experimental work 2) inferior model throughput because of extra computations and additional usage of memory.
> Better wording will be used in the final version (currently lines 27-32 provide sufficient context in the main text).
>
> About T5. We believe that because SOTA relative positional embeddings (learnable or sinusoidal) are too expensive for large-scale experiments (see for example paper "William Chan, Daniel Park, Chris Lee, Yu Zhang, Quoc Le, and Mohammad Norouzi. Speech-stew: Simply mix all available speech recognition data to train one large neural network. arXiv
> preprint arXiv:2104.02133, 2021." where they stick with absolute sinpos for large-scale experiments mentioning about expensive relative positional embedding), for T5 they did simplification to speed up the training (also they wrote that "for efficiency, we also share the position embedding parameters across all layers in our mode"). From T5 paper it is unclear how much quality deteriorated because of this simplification and further experimental study is needed for their version of relative positional embedding (from T5 paper: "Since these architectural changes are orthogonal to the experimental factors we consider in our empirical survey of transfer learning, we leave the ablation of their impact for future work"). Performance of T5 on En-Fr task (with a dramatic amount of external data) is not convincing.
>
> > When discussing complex-valued representation of embeddings in line 81. Please consider the following reference: Wang et.al. encoding word order in complex embeddings. ICLR 2020. which also explored the continuous nature of position embeddings.
>
> Thanks for pointing to this reference, it will be included in the final text.
>
> > minor issue: Figure 1 was not mentioned anywhere in the main text.
>
> Thanks for catching this, we will add the reference in the text.

---

### Official Review · Reviewer_iFjJ · 2021-07-16

**Rating:** 8
**Confidence:** 3

**Summary:**

This paper proposes “CAPE” (continuous augmented positional embeddings), a novel form of positional embeddings, aimed at making transformer models more robust to changes in the input distribution at inference time (e.g., changes in input size/length/resolution, changes in relative distances/positions between subunits of input, and changes in absolute positions of input subunits).  It accomplishes this by (1) indexing positional embeddings in a continuous way (e.g., for images: indexing by the normalized coordinate $(x,y)  \in [-1,+1]^2$ of a patch; for audio/speech: indexing by the time (in seconds) of a frame), and (2) performing a series of augmentations (global shift, local shift, global scaling) to these “continuous indexes” at each position in the input (these augmentations can be thought of as being similar to performing augmentations on the input).

The paper presents experimental results across image classification (vision), ASR (speech), and machine translation (NLP) tasks, showing that in general, the proposed CAPE embeddings outperform baseline positional embedding methods. In particular, CAPE is compared with (1) _nopos_: no positional embeddings, (2) _abspos_: learnable absolute positional embeddings, (3) _sinpos_: continuous sinusoidal positional embeddings (without the proposed augmentations), and (4) _relpos_: learnable relative positional embeddings (for ASR tasks).


**Detailed summary of experimental results, by domain**

_Image Classification_: The experiments demonstrate that ViT models trained with CAPE embeddings perform better than the baseline embedding methods across test images of different resolutions. Furthermore, the paper proposes “UniViT”, a ViT model trained with CAPE embeddings and randomly resized inputs (128x128 -> 320x320), and shows that UniViT performs extremely well across test image resolutions (160x160 -> 672x672) relative to ViT models trained only on a single resolution.  Because the UniViT model gracefully handles images of different resolutions, this model also provides a way for practitioners to easily trade-off increased throughput for lower accuracy, by lowering the resolution of the input images fed to the model at inference time.

_ASR_: The paper shows that CAPE generally outperforms the other absolute positional embeddings (nopos, sinpos, abspos), while generally being competitive with, or outperforming, the more computationally expensive relative positional embeddings (relpos). It additionally shows that (1) CAPE performs similarly to using SpecAugment as a form of data augmentation during training (and combining CAPE + SpecAugment does best), and (2) CAPE can be used, together with adjusting the STFT hop distance of utterances in a batch, to eliminate the need for padding during ASR training (without degrading performance).

_Machine Translation_:  The paper again demonstrates strong performance for CAPE.


**Limitations And Societal Impact:**

I think a broader discussion about the limitations would be helpful (only 1 vacuous sentence was included after the conclusion). For example, discussing when relative positional embeddings would attain better performance than any absolute positional embeddings (e.g., CAPE). From a representation perspective, what can’t be captured by CAPE embeddings?

In terms of societal impact, the authors could argue that these embeddings provide a relatively simple way of producing more efficient transformer models (by downsampling input, at least for images/audio), and perhaps mention how CAPE could be combined with other efficient transformer innovations (e.g., performer / other “linear attention” transformers).


**Main Review:**

### Originality
The work is quite original in my opinion. In particular, the idea of adding a series of augmentations to the positional embeddings, in order to make the model more robust to shifts/distortions in the input at inference time, is a simple/powerful/novel idea. The way in which CAPE embeddings are applied to image classification, to train a model which is robust to different image resolutions, and thus allow for practitioners to perform faster/cheaper inference by downsampling input images, is also quite original.

### Quality
The quality of the work is high. The paper presents thorough experiments across three different input domains (images, speech, text), comparing throughout with important baselines.

### Significance
I believe this paper has the potential to be relatively impactful, for the following reasons.
1) It offers a domain-agnostic framework for performing augmentation for transformer models, yielding more robust (and higher accuracy) models (e.g., UniViT). For example, I believe it is in many cases simpler to use CAPE than to design domain-specific augmentations (e.g., SpecAugment). CAPE can also be combined with domain-specific augmentations in order to yield even better performance.
2) The continuous indexing proposed by CAPE allows for training models which are robust to inputs of different lengths, as well as to stretching/compressing inputs.  As a result, CAPE offers an easy way to either (a) make inference cheaper by downsampling inputs, or (b) get rid of padding in ASR training by adjusting STFT hops.

### Clarity
In general, the paper is written in a very clear way.  Some questions about specific points meriting further clarification are included at the end of this review.

Throughout the paper, I think it would be beneficial to be more specific about the performance differences between CAPE and the baseline methods (e.g., CAPE performs XX% absolute better/faster/?? than YY on ZZ)---this would help clarify to the reader the exact gains from CAPE, without them needing to zoom into specific parts of a figure.  These numbers could be included in the abstract/introduction.

## Overall
I think this paper would make a great contribution to NeurIPS, and I recommend acceptance for this work.

**Main strengths**:
* I think the accuracy improvements from using CAPE in numerous domains (especially in different-from-training settings), the UniViT model which performs well across image resolutions, the ability to easily navigate the accuracy-computation trade-off by modifying input size (e.g., Figure 4), and the general domain-agnostic framework for data augmentation introduced by CAPE, all have the potential to be impactful.

**Main weaknesses**:
* I think the main weaknesses of the work are clarity in a few sections (see below), an incomplete empirical comparison with (and related work discussion about) existing methods for dealing with images of different resolutions, and a way-too-brief discussion of the limitations of CAPE.

Importantly, I am not an expert in the latest literature about positional embeddings for transformers, and thus my review is only medium-confidence.

I now provide a longer/more detailed list of questions for the authors.

### Clarifications/Questions:
1) Choices of equations for positional embeddings:
* It was unclear to me how the equations for $w_{k,x}$ and $w_{k,y}$ were chosen for the positional embeddings for images.  In particular, why does $w_{k,x}$ include a multiplication by $\cos(k)$, and $w_{k,y}$ include a multiplication by $sin(k)$?  Why are $w_{k,x} x$ and $w_{k,x} y$, combined additively (for example, as opposed to computing positional embeddings in the x and y dimensions separately, and concatenating)?  Have fixed (as opposed to learned) positional embeddings been used for image classification before, and if so, what functional forms were chosen for those positional embeddings?
* Across the three domains, how were the different scaling terms for $w_k$ chosen ($10000^{-2k/K}$, $10^{2k/K}$, $30 \cdot 10^{-2k/K}$)?

2) Image classification experiments:
* It was not clear to me if when you train (or test) on images of different resolutions, this corresponds to (a) training (or testing) on completely disjoint sets of images, whose original resolutions match the stated resolution (e.g., when you train with 224x224 images, this corresponds to training on all images in the ImageNet dataset whose native resolution is 224x224), or (b) training (or testing) on the same set of images, which have been adjusted (scaled/cropped) to match the stated resolution (e.g., take all the ImageNet images, and rescale them to be 224x224).  Can you please clarify this? For example, is UniViT trained on a larger set of images than the resolution-specific ViT models (or on the same set of images, but with random rescaling)?  On a related note: a discussion about the ImageNet dataset, the resolutions of the images in this dataset, and the way these differences in resolution have been handled by prior work (e.g., via cropping), would be quite helpful.
* What, in your opinion, is the difference between (1) training a ViT model with CAPE on images that have all been scaled to a certain resolution (e.g., 224x224), (2) training using CAPE on images of different resolutions, and (3) training on randomly resized images (e.g., UniViT)? It seems that all of these approaches would nicely handle images of different sizes (although the ViT models which take as input images of different resolutions would be cheaper on low-resolution images, and more expensive on high-resolution images).  Could comparisons between these different approaches be included in the paper?

3) ASR experiments:
* Why is the proposed method for getting rid of padding (STFT hop distance tuning), better than simply doing padding? It doesn’t seem that the proposed method is simpler than padding, more computationally efficient than padding, or better performing than padding (dashed and solid lines in figures 15 and 16 seem to perform quite similarly). I think it would be helpful to clarify these points in the main text.
* Could an equivalent of Figure 4 be produced for ASR?  For example, plot WER of utterances as you increase the STFT hop distance?
* Could an equivalent of UniViT be trained for ASR?  For example, training a model on randomly chosen STFT hop distance?

4) MT experiments:
* Could an equivalent of Figure 4 be produced for MT? For example, plotting BLEU as you drop the least meaningful words in the input sentence? At first glance, this seems quite difficult, but I was curious if this accuracy-computation trade-off exists in the NLP domain as well. For example, perhaps you can still generate good translations after dropping stop words? Perhaps this would be easier to demonstrate on sentence classification tasks, in which ignoring certain words in the input might not make a big impact in the ability of the model to classify the sentence correctly (though one would have to be careful with this pruning, since removing certain words (e.g., negations) could completely change the correct answer).


**Time Spent Reviewing:**

5.5

---

> ### Author Response · Authors · 2021-08-10
> **First comments/explanations**
>
> We deeply appreciate and sincerely thank anonymous reviewer iFjJ for detailed review of our paper (even including appendix). These  detailed comments help us further improve the clarity of our paper. Please find our detailed comments and explanations below.
>
> > Throughout the paper, I think it would be beneficial to be more specific about the performance differences between CAPE and the baseline methods (e.g., CAPE performs XX% absolute better/faster/?? than YY on ZZ)---this would help clarify to the reader the exact gains from CAPE, without them needing to zoom into specific parts of a figure. These numbers could be included in the abstract/introduction.
>
> Thanks for this important remark, we'll try to integrate this practice into our writing!
>
> > It was unclear to me how the equations for $w_{k,x}$ and $w_{k,y}$ were chosen for the positional embeddings for images. In particular, why does $w_{k,x}$ include a multiplication by $\cos(x)$, and $w_{k,y}$ include a multiplication by $\sin(k)$? Why are $w_{k,x}$ and $w_{k,y}$, combined additively (for example, as opposed to computing positional embeddings in the x and y dimensions separately, and concatenating)? Have fixed (as opposed to learned) positional embeddings been used for image classification before, and if so, what functional forms were chosen for those positional embeddings?
>
> Figure 13 in appendix should be helpful: it visualizes embedding from abspos, sin2d and CAPE. "inner" $\sin k$ and $\cos k$ in the formula are responsible for angle of "hatching" (so all components have different angle, angles uniformly cover possible directions), while "outer" sin and cos used in definition of $E_{2k}$ and $E_{2k+1}$ create a wave as in 1d sinusoidal embedding and control hatching density. This choice was made because of following considerations: a) simple-to-reproduce implementation b) no "selected directions" on the plane, when compared to combination of 1d embeddings c) embedding is deterministic, no RNG used d) different "hatching densities" allow both precise and approximate positioning.  Additionally, compared to concatenation of 1d embeddings, proposed schema allows attending to specific small region around a point on image without emphasizing points with only same x or only same y - while we expect this artifact to contribute with 1d embeddings.
>
> Please refer to the concurrent work https://arxiv.org/pdf/2102.10882.pdf we cite: authors perform an ablation on fixed positional embedding (sinusoidal) with concatenation of embeddings for different axes. Prior works also used the same strategy: sinusoidal positional embedding + concatenation for several axes: https://arxiv.org/pdf/1802.05751.pdf, https://openaccess.thecvf.com/content_ICCV_2019/papers/Bello_Attention_Augmented_Convolutional_Networks_ICCV_2019_paper.pdf. In the original ViT paper https://arxiv.org/abs/2010.11929 authors performed ablations on positional embedding: nopos, abspos 1D, abspos 2D (where for each axis x and y separate D/2 embeddings are learned and then concatenated - the same way as for sinpos), variations where to add them. They did not observe any improvement by encoding coordinates separately.
>
> > Across the three domains, how were the different scaling terms for $w_{k}$ chosen ($10000^{-2k/K}$, $10^{-2k/K}$, $30\cdot 10^{-2k/K}$ )?
>
> For the original vanilla transformer 10000 is chosen to provide distinct frequencies for positions up to 10000 which is enough in case of modeling text (we just follow common practiсe here). For other modalities two main factors contribute: how precise the location should be for encoding (e.g. there is no need to encode position in audio much more precisely than one syllable takes), second is longest reasonable "distance" that is expected in the data. Second factor e.g. was set in the original transformer paper as 10k tokens (hence normalization), we also tried to have some reasonable margin on both precision and "length" of the parameter.
> We can reformulate CAPE in terms of these values, but we believe that for all practical cases provided choices are reasonable for practitioners.
>
> **Image classification experiments:**
>
> > It was not clear to me if when you train (or test) on images of different resolutions, this corresponds to (a) training (or testing) on completely disjoint sets of images, whose original resolutions match the stated resolution (e.g., when you train with 224x224 images, this corresponds to training on all images in the ImageNet dataset whose native resolution is 224x224), or (b) training (or testing) on the same set of images, which have been adjusted (scaled/cropped) to match the stated resolution (e.g., take all the ImageNet images, and rescale them to be 224x224). Can you please clarify this? For example, is UniViT trained on a larger set of images than the resolution-specific ViT models (or on the same set of images, but with random rescaling)? On a related note: a discussion about the ImageNet dataset, the resolutions of the images in this dataset, and the way these differences in resolution have been handled by prior work (e.g., via cropping), would be quite helpful.
>
> Our training strategy is (b) - the same set of images is used in all experiments. ImageNet dataset has images of a variety resolution, even min(h, w)=5000 for some images. Some statistics on image resolutions can be found in Table 5 in the appendix B.2. About prior work to handle different resolutions, see ViT and specifically DeiT (they also study how fine-tuning on higher resolution influences) papers.
>
> > What, in your opinion, is the difference between (1) training a ViT model with CAPE on images that have all been scaled to a certain resolution (e.g., 224x224), (2) training using CAPE on images of different resolutions, and (3) training on randomly resized images (e.g., UniViT)? It seems that all of these approaches would nicely handle images of different sizes (although the ViT models which take as input images of different resolutions would be cheaper on low-resolution images, and more expensive on high-resolution images). Could comparisons between these different approaches be included in the paper?
>
> Yes, all the cases can be applied to any resolution and as we demonstrated even ViT with CAPE has better performance on the resolutions different from train one than vanilla ViT.
> In the paper, with Figure 2, we compare (1) and (3). We believe that (3) is more powerful because every image would appear during training in different resolutions. Some caveat in implementing (2) in experiments is that the vast majority of images have resolution >> 600x600, so there is no 'natural size' for them.
>
> **ASR experiments**:
>
> > Why is the proposed method for getting rid of padding (STFT hop distance tuning), better than simply doing padding? It doesn’t seem that the proposed method is simpler than padding, more computationally efficient than padding, or better performing than padding (dashed and solid lines in figures 15 and 16 seem to perform quite similarly). I think it would be helpful to clarify these points in the main text.
>
> Simplification we point to is in the implementation (of model, and implementation of building blocks of deep learning framework). Examples: normalization (any!) should be aware which tokens are "pad-tokens", otherwise normalization constants would depend on the amount of padding; any attention module should be aware which tokens it should not attend to, and for seq2seq models this is critical.
> In case of a padding-free approach there is no padding-token at any level in the network, and a lot of implementational caveats with propagating padding positions are removed.
> Efficiency argument refers to efficiency of training. High spread in lengths of audio (e.g. 2s and 15s) forces practitioners to sort utterances by length to minimize padding: single long utterance drives to large amount of padding for all short utterances in a batch. This reduces variability in batches drastically. We demonstrate that with CAPE and pad-free pipeline we can 1) mix utterances of not-too-different lengths within a batch, so better randomization 2) utilize all tokens 3) achieve small improvements (figures 15-16), improvements on in-house data are quite consistent.
>
> > Could an equivalent of Figure 4 be produced for ASR? For example, plot WER of utterances as you increase the STFT hop distance?
>
> Thanks for this idea, we will implement this benchmarking.
>
> > Could an equivalent of UniViT be trained for ASR? For example, training a model on randomly chosen STFT hop distance?
>
> STFT hop distance augmentation and UniViT are somewhat parallel realizations of the same idea, but each adapted for the domain. One can think of changing STFT distance as a way to "resize" audio data and adjust ASR throughput, however very low resolution (like 128x128) in images works well enough while speech granularity is far more sensitive to "resizing" as phoneme duration can take as little as 30-50ms.

---

> > ### Author Response · Authors · 2021-08-10
> > **First comments/explanations-2**
> >
> > **MT experiments:**
> >
> > > Could an equivalent of Figure 4 be produced for MT? For example, plotting BLEU as you drop the least meaningful words in the input sentence? At first glance, this seems quite difficult, but I was curious if this accuracy-computation trade-off exists in the NLP domain as well. For example, perhaps you can still generate good translations after dropping stop words? Perhaps this would be easier to demonstrate on sentence classification tasks, in which ignoring certain words in the input might not make a big impact in the ability of the model to classify the sentence correctly (though one would have to be careful with this pruning, since removing certain words (e.g., negations) could completely change the correct answer).
> >
> > That's an interesting idea and it crossed our minds, but MT is certainly an inappropriate task. We are also not aware of cheap approaches to "minimizing sentence length" to provide some valuable study in that direction. It also doesn't look like modern NLP is concerned much about resources :)
> >
> > > I think a broader discussion about the limitations would be helpful (only 1 vacuous sentence was included after the conclusion). For example, discussing when relative positional embeddings would attain better performance than any absolute positional embeddings (e.g., CAPE). From a representation perspective, what can’t be captured by CAPE embeddings?
> >
> > From a representation perspective we demonstrate CAPE is capable of providing relative shifts. However, a model should "learn" this relative positioning. Thus we can expect that relpos should be beneficial in settings with small amounts of data because of more appropriate inductive bias. We'll try to include that in the discussion.
> >
> > > In terms of societal impact, the authors could argue that these embeddings provide a relatively simple way of producing more efficient transformer models (by downsampling input, at least for images/audio), and perhaps mention how CAPE could be combined with other efficient transformer innovations (e.g., performer / other “linear attention” transformers).
> >
> > We are discussing this a bit in the conclusion section, rows 293-297.
> > Thank you for pointing out the "linear attention" transformers. CAPE can be readily plugged in there (contrary to relative positional embedding). We will work on adding your suggestions for the final version.

---

### Author Response · Authors · 2021-08-23
**Additional ablations/benchmarks asked by reviewers.**

Dear reviewers,

We have performed additional experiments to cover previously asked questions. This should complete the picture of CAPE.

- First is benchmarking the WER vs throughput for ASR models when varying the STFT hop distance in the data processing. The plot can be found via this link https://ibb.co/0FPJy60 (for TED-LIUM models on TED-LIUM test set). As anticipated, if a model is trained with augmentation on STFT hop distance it's WER is less affected by varying STFT hop distance. This proves possibility for on-the-fly throughput adjustments in ASR, while not as impressive as in image recognition.

- Second, we added experiments for MT with learnable relative positional embedding for 6L-6L configuration both for En-Fr and En-De benchmarks. In this experiment the positional embedding has a context size of 150 tokens to the left/right to cover training sequences. Also we added a CAPE model with a larger global shift, +/- 50. In terms of BLUE score results are similar to ASR: relpos and CAPE are in the same ballpark. We further compare generalization to longer sequences via the plot of logloss per position (https://ibb.co/9HKptWX), where all embeddings outperform relpos for positions < 100 while on positions > 200 relpos outperforms others. For the En-Fr benchmark (which we found to be more stable due to larger training and validation data), CAPE with the 50 global shift behaves similar to the relpos for very large positions being better for first positions. This allows us to conclude that CAPE features advantages of both absolute positional embeddings and relative positional embeddings for the MT task.

| Model | Lang | valid BLUE| test BLUE |
| - | - | - | - |
sinpos, 6L-6L | DE | 26.88 +/- 0.05 | 27.66 +/- 0.10
abspos, 6L-6L | DE | 26.68 +/- 0.05 | 27.36 +/- 0.06
relpos, 6L-6L | DE | 26.81 +/- 0.16 | 27.92 +/- 0.07
CAPE, 6L-6L, $\Delta=5$ | DE | 26.86 +/- 0.13 | 27.89 +/- 0.07
CAPE, 6L-6L, $\Delta=50$ | DE | 27.09 +/- 0.03 | 27.77 +/- 0.16
|  | |  |  |
sinpos, 6L-6L | FR | 47.27 +/- 0.03 | 41.13 +/- 0.07
abspos, 6L-6L | FR | 47.22 +/- 0.03 | 41.21 +/- 0.04
relpos, 6L-6L | FR | 47.12 +/- 0.03 | 41.32 +/- 0.13
CAPE, 6L-6L, $\Delta=5$ | FR | 47.22 +/- 0.03 | 41.59 +/- 0.03
CAPE, 6L-6L, $\Delta=50$ | FR | 47.14 +/- 0.02 | 41.48 +/- 0.10

- Third, we benchmarked the speed reduction for learnable relative positional embedding: we measure average time over 100 runs for forward+backward for a) multihead attention block b) encoder block c) decoder block with relative positional embedding; batch is set to 50, embedding dimension is 768 and number of heads 8 (one of the standard settings for transformer block); we apply to the input sequences of length 10, 100, and 1000 performing computations in fp32 or fp16 on the GPU V100 32Gb; relative positional embedding uses either 100 or 1000 context to left/right. In the table below we report the slowdown for relpos (time(with relpos) / time(without relpos)).

| Model | FP16 Len-10 | FP32 Len-10 | FP16 Len-100 | FP32 Len-100 | FP16 Len-1000 | FP32 Len-1000 |
| - | - | - | - | - | - | - |
Block, context 1000 | 2.1 | 2.3 | 3.3 | 2.3 | 2.2 | 1.7
Encoder, context 1000 | 2.2 | 2.2 | 2.6 | 1.7 | 2.0 | 1.6
Decoder, context 1000 | 1.8 | 1.7 | 1.9 | 1.4 | 1.6 | 1.4
Block, context 100 | 2.1 | 2.1 | 1.9 | 1.2 | 1.4 | 1.2
Encoder, context 100 | 2.3 | 2.4 | 1.5 | 1.1 | 1.4 | 1.2
Decoder, context 100 | 1.7 | 1.7 | 1.2 | 1.1 | 1.2 | 1.1

---

> ### Comment · Area_Chair_KYko · 2021-08-27
> **Responses**
>
> Thanks the authors for providing additional experimental results!
>
> The AC has a few questions about the proposed techniques. In tasks such as object detection (https://arxiv.org/abs/2005.12872), pixel positions are normalized between 0 and 1. In language models such as Universal Transformer (https://arxiv.org/pdf/1807.03819.pdf), global random shift has also been used to avoid the model memorizing positions. Can the authors further clarify the novelty of the proposed techniques in light of these previous techniques? In addition, is there an ablation to show how each augmentation contributes to the overall performance?

---

> > ### Author Response · Authors · 2021-08-27
> > **Clarification on related works and ablation study**
> >
> > Dear AC,
> >
> > Thanks a lot for the comments and important questions! Please find below our answers:
> >
> > >  In tasks such as object detection (https://arxiv.org/abs/2005.12872), pixel positions are normalized between 0 and 1.
> >
> > - Normalization is not a novelty, but a reasonable choice to avoid extrapolation when resolution is changed
> > - To our knowledge, neither previous work used position augmentations during positional encoding in vision and speech
> > - We further introduced the first deterministic positional encoding for images (2d) that has no "selected directions" on the plane, see formula in lines 90-91
> > - We are not aware about previous works on transformers that reliably perform on images of multiple resolutions
> >
> > >  In language models such as Universal Transformer (https://arxiv.org/pdf/1807.03819.pdf), global random shift has also been used to avoid the model memorizing positions. Can the authors further clarify the novelty of the proposed techniques in light of these previous techniques?
> >
> > Many thanks for pointing to this work, we will properly reflect their contribution in the paper.
> > In the UT paper, they used discrete global shifts in one group of experiments (algorithms), where inability to generalize to longer sequences critically hits the performance. However, performance without shifts (ablation) is not reported.
> > Below we focus on our contribution on the top of UT work:
> > - We use continuous augmentations, not discrete. Our augmentations include not only global shift (offset in UT terms), but also local shift and global scaling
> > - Continuous augmentations are more natural for continuous modalities like images and speech, and CAPE shows largest benefits for continuous modalities, not discussed in UT
> > - For machine translation we don't use an approach from UT paper, rather we do synchronized global shift and scale in positions in both source and target languages: according to their published code, they introduce independent shifts multiple times throughout the model (we do that once for encoder and once for decoder and in synchronized manner). Thus we don't overlap in experiment settings and method with UT, but have an aligned vision of overcoming generalization issue
> > - We discuss the role of additional augmentations: in supplementary (figure13) we demonstrate that local shift augmentation destroys Moire-like patterns (which persist under the global shift); global scaling prevents memorization of relative positions (relative positions also persist under the global shift)
> > - Finally, a switch to continuous positions and augmentations we made in this work allows new pipelines like UniViT and STFT hop distance augmentation, because these pipelines rely on varying spatial density of tokens
> >
> > > In addition, is there an ablation to show how each augmentation contributes to the overall performance?
> >
> > We have performed these ablations, however they can't be squeezed in the main text. Please find ablation experiments and their discussion in the supplementary material:
> > - In section B.3 for image recognition:
> >   - Fig. 10 how global scaling influences the models trained on different resolutions
> >   - Fig. 11 all combinations on usage of the augmentations: only one is used, only pair is used, all three are in use. Global scaling is the most important to have better generalization ability.
> >   - Fig. 12 how global scaling influences the UniViT (better to set scaling to 1 as training happens now on a mix of resolutions).
> > - In section C.4 for speech recognition:
> >   - Fig. 17 how global shift value influences the generalization ability (overall larger value is better).
> >   - Fig. 18 how local shift influences performance and generalization.
> >   - Fig. 19, 20 how global scaling influences performance and generalization.
> > - Recent comment on MT compares performance and generalization when global scaling is varied
> >
> > Contribution of specific augmentations varies between modalities, which is expectable: the same augmentations (crop/resize/pitch shift/echo addition) are rarely used across modalities.
> >
> > Let us know if you have other questions or concerns.

---

> > > ### Comment · Area_Chair_KYko · 2021-09-01
> > > **Responses**
> > >
> > > Thank you for your clarifications.

---

### Decision · Program_Chairs · 2021-09-27

**Decision:**

Accept (Poster)

**Comment:**

Although there is some variance in the reviewer assessments about the paper, most reviewers are positive and think the work nicely integrates a set of techniques for positional encoding that can capture relative positional relationships. The work evaluates these techniques in several distinct tasks, including vision, language and speech. As Reviewer MLDf commented: "It (the technique) seems to help on speech and vision, but not so much on machine translation." I would want to thank the authors for providing extensive additional experimental results in the rebuttal, which strengthen the paper. The reviewers have provided many great points for improving the paper, which I hope the authors can include them in the revision of the paper.